

**Sources and trends of Black Carbon Aerosol in a Megacity of Nanjing, East China After the China Clean Action Plan and Three-Year Action Plan**

*Abudurexiati·Abulimiti [a,b], Yanlin Zhang [a,b*], Mingyuan Yu [a,b], Yihang Hong [a,b], Yu-Chi Lin [a,b],*

*Chaman Gul [c], Fang Cao [a,b]*

[a] *School of Ecology and Applied Meteorology, Nanjing University of Information Science and Technology, Nanjing, 210044, China*

[b] *Atmospheric Environmental Center, Joint Laboratory for International Cooperation on Climate and Environmental Change, Ministry of Education, Nanjing University of Information Science and Technology, Nanjing, 210044, China*

[c] *Reading Academy, Nanjing University of Information Science and Technology, Nanjing, Jiangsu, 210044, China*

*Correspondence to:*

Yanlin Zhang* (zhangyanlin@nuist.edu.cn, dryanlinzhang@outlook.com)

**Abstract** Black carbon (BC) is an essential component of particulate matter (PM) with a significant impact on climate change. Few studies have investigated the long-term changes in BC and the sources, particularly considering primary emissions of BC, which is crucial for developing effective mitigation strategies. Here, based on three-year observations (2019-2021), random forest (RF) algorithms were employed to reconstruct BC concentrations in Nanjing from 2014 to 2021. Source apportionment was conducted on the reconstructed data to investigate long-term trends of BC and its sources. The results showed that the three-year average BC concentration was $2.5\pm1.6$ $\mu g$ $m^{-3}$, peaking in winter, with approximately 80% attributed to liquid fuel combustion. Notably, the reconstructed time series revealed a significant decrease ($p < 0.05$) in BC levels over the eight-year period, primarily due to reduced emissions from liquid fuels. The comparison between two control polices periods (P1:2014-2017 and P2:2018-2021) indicate that BC concentrations decline more steeply during S2 since significant ($p < 0.05$) reduction in biomass burning. The seasonal analysis showed significant reductions ($p < 0.05$) in BC, $BC_{liquid}$ (black carbon from liquid fuel combustion) and $BC_{solid}$ (black carbon from solid fuel combustion) during winter, with $BC_{liquid}$ accounting for 77% of the reduction. Overall, emission reduction was the dominant factor in reducing BC levels, contributing between 62% and 86%, as revealed by Kolmogorov-Zurbenko



(KZ) filter. However, during P2, meteorological conditions played a more
significant role, especially in reducing BC and BC$_{liquid}$, with an increase in
their impact on BC$_{solid}$ compared to P1. Our results demonstrated that target
control measures for liquid fuel combustion are necessary, as liquid fuel
combustion is a major driver of decreasing BC, especially in summer, while
the influence of meteorological factors on BC variations cannot be overlooked.
**Keywords:** black carbon; sources; random forest; emission reduction
**1. Introduction**
Black carbon (BC), also known as element carbon (EC), is a
carbonaceous component of particulate matter (PM) produced through
incomplete combustion processes, including domestic cooking, heating and
coke-making (Bond et al., 2013; Liu et al., 2020). BC particles significantly
influence the Earth's energy balance and are major contributors to global
warming due to their strong absorption of solar radiation across visible to
infrared wavelengths (Ramanathan and Carmichael, 2008; Ipcc, 2023).
Additionally, the presence of BC particles in the atmosphere reduces
atmospheric visibility and deteriorates air quality especially in urban areas
due to their significant absorption properties (Ding et al., 2016). Exposure to
BC aerosols has also been linked to increased health risks, such as heart
attacks and cardiovascular diseases (Sarigiannis et al., 2015; Li et al., 2019).
Owing to its short atmospheric lifetime of only 3 to 14 days, much shorter
than that of greenhouse gases which can persist for decades, reducing BC
emissions can promptly mitigate global warming and benefit human health.
Accurate quantification of BC from different sources is essential to
propose efficient mitigation strategies. Various methods in the past have been
applied to BC source apportionment, including emission inventories (Zhu et
al., 2020), radiocarbon isotope analysis (Zhang et al., 2014; Yu et al., 2023),
and receptor models (Zong et al., 2016). However, uncertainties arise due to
the lack of reliable emission factors, and receptor models require additional
aerosol composition data. The radiocarbon source apportionment method is
limited by its low temporal resolution, which hinders their ability to capture
the dynamic changes in BC sources. In contrast, the Aethalometer model,
with its high temporal resolution and rapid analysis, has been widely adopted
for quantifying BC derived from liquid fuel (BC$_{liquid}$) and solid fuel (BC$_{solid}$)
combustion (Lin et al., 2021; Sandradewi et al., 2008; Helin et al., 2018).



To address the sever air pollution issue, the Chinese government implemented the "China Clean Action Plan" during 2013-2017 and the "Three-Year Action Plan" during 2018-2020. Several studies in recent years have focused on long-term BC mass concentrations in major cities or regions of China to evaluate the impact of emission reduction measures implemented by the Chinese government (Sun et al., 2022a; He et al., 2023). However, while most of these studies document changes in BC concentrations, few have explored the specific contributions of different BC sources. Such an understanding is essential for identifying the drivers behind observed changes and for developing targeted mitigation strategies. Moreover, comprehensive datasets of BC are crucial for a better understanding of BC mass concentration variations and their implications for air quality policy. However, newly established monitoring stations often lack sufficient long-term observations, making it difficult to evaluate historical variations in BC concentrations. This limitation hinders efforts to understand BC dynamics in regions with limited prior monitoring, ultimately complicating the formulation of effective emission reduction policies. Chemical transport models (CTMs), which integrate meteorological conditions and emission inventories, are effective in simulating near-surface BC concentrations over short term periods (Cheng et al., 2019; Zhou et al., 2023). Nonetheless, their computational intensity and time-consuming often limit their application to long-term simulation. In contrast, the prediction of PM or other air pollutants can be efficiently achieved through statistical models that establish relationships between measured values and various variables, including co-emitted pollutants, air humidity and air temperature. Recently, the historical values of nitrate $\delta^{15}N$ and $PM_{2.5}$ have been accurately reproduced based on the statistical relationships established between measured variables and other influencing factors (Fan et al., 2023; Zhao et al., 2020; Wu et al., 2024). This method provides a relatively straightforward approach for simulating historical air pollutants and is accurate enough for examining their long-term variations.

The long-term variation of atmospheric aerosol composition can be attributed to both meteorology conditions and emissions. CTMs are one of the often used tools to quantify the impact of meteorology and emission on aerosols, as they consider the physical and chemical process that air pollutants undergo during their time in the atmosphere (Li et al., 2023; Zhang et al., 2019;



Du et al., 2022). However, the accuracy of CTMs is often constrained by their
initial conditions and uncertainty in emission inventory as well as models'
underlying assumptions. Another commonly used method for separating the
influences meteorology and emissions on target atmospheric pollutants is the
Kolmogorov-Zurbenko (KZ) filter. For example, Sun et al. (2022b) found that
meteorological contribution to the $PM_{2.5}$ trend presented a distinct spatial
pattern over the Twain-Hu Basin, with northern positive rates up to 61% and
southern negative rates down to -25%. Chen et al. (2019) reported that
anthropogenic emissions contributed to 80% of reduction in $PM_{2.5}$ in Beijing
from 2013 to 2017. Compared to CTMs, the KZ filter is easier to operate and
is suitable for any long-term datasets of air pollutants, making it a practical
tool for analyzing trends in atmospheric pollutants.
In the present study, a three-year BC mass concentration measurement
was conducted to clarify BC characteristics and quantify contributions from
different sources. The measured BC at two wavelengths (370nm and 880 nm)
then incorporated into random forest model to establish the nonlinear
relationships with predictor variables, such as air pollutants and
meteorological factors. Historical BC concentrations at the two wavelengths
were reconstructed from 2014-2021 using the trained models to investigate
the long-term temporal variation of BC and sources, with a focus on the two
distinct emission reduction periods: the "China Clean Action Plan" and the
"Three-Year Action Plan". Finally, the impacts of meteorology and emissions
on the long-term trend of BC were quantified to provide deeper insights into
the factors driving its historical changes.
**2. Data and Methods**
**2.1 Sampling site and Data**
Nanjing is located eastern part of China, is vital industrial and economic
center. The sampling instrument used for monitoring BC mass concentration
was positioned on the rooftop of a seven-story building at the campus of
Nanjing Information Science and Technology (NUSIT, 32.21°N, 118.72°E,
Figure S1 in Supporting Information), Nanjing, China. The sampling site
represents a typical urban atmospheric environment, encircled by local roads
with an expressway approximately 1 km away. Moreover, a steel
manufacturing plant and a petroleum chemical factory were about 5 km away
from the sampling site. Traffic and industrial emissions are the primary





sources of air pollution at the sampling site. Nanjing experiences four
dominant seasons each year: winter (December-February), spring (March-
May), summer (June-August), and autumn (September-November).
A dual-spot Aethalometer (AE33, Magee Scientific) was used to
measure BC mass concentration from January 2019 to December 2021. The
flow rate of AE33 was set to 5 L min$^{-1}$ and the inlet cut-off size was 2.5 μm
throughout the entire period. In brief, aerosol particles were collected on a
filter tape automatically, and light attenuations (ATN) were measured at seven
distinct spectral regions (370, 470, 520, 590, 660, 880, 950 nm) with a time
resolution of 1 min. The ATNs were then converted to BC mass
concentrations with seven different mass absorption cross sections (18.47,
14.54, 13.14, 11.58, 10.35, 7.77, 7.19 m$^2$ g$^{-1}$). In this study the BC
concentration calculated by 880 nm spectral region was used, as BC is the
predominant absorber at this wavelength (Drinovec et al., 2015). The BC data
was missing since instrument maintenance from 13$^{th}$ July to 31$^{st}$, 2020, and
from July 23$^{rd}$ to September 26$^{th}$, 2021. Hourly averaged concentrations of
PM$_{2.5}$, CO, SO$_2$ and NO$_2$ were obtained from the China National Air Quality
Monitoring Station, located approximately 10 km from the sampling site.
Hourly resolution meteorological data, including temperature (T), relative
humidity (RH), wind speed (WS), wind direction (WD) and boundary layer
height (BLH), were sourced from the ERA5 reanalysis datasets provided by
the European Centre for Medium-Range Weather Forecasts (ECMWF).
**2.2 Aethalometer measurements and source apportionment**
The absorption Ångström exponent (AAE) describes the spectral
dependence of BC and is determined through a power-law fit between light
absorption (b$_{abs}$(λ)) and seven wavelengths, the equation can be written as:

$$b_{abs}(\lambda) = \beta \cdot \lambda^{-AAE} \qquad (1)$$

where β is a constant dependent on aerosol mass concentration and size
distribution. Subsequently, the Aethalometer model is utilized to quantify the
contribution of liquid and solid fuels to BC. The model assumes that ambient
BC primarily originates from liquid fuel and solid fuel combustion, with BC
from two distinct combustion sources having differing light absorption
spectra. Hence, the total light absorption at 880 nm is attributed to liquid fuel-
generated BC (BC$_{liquid}$) and solid fuel-derived BC (BC$_{solid}$). The relationships



between $b_{abs}(\lambda)$, $\lambda$ and AAE can thus be expressed as follows:

$$\frac{b_{abs}(\lambda_1)_{liquid}}{b_{abs}(\lambda_2)_{liquid}} = \left(\frac{\lambda_1}{\lambda_2}\right)^{-AAE_{liquid}} \tag{2}$$


$$\frac{b_{abs}(\lambda_1)_{solid}}{b_{abs}(\lambda_2)_{solid}} = \left(\frac{\lambda_1}{\lambda_2}\right)^{-AAE_{solid}} \tag{3}$$


$$b_{abs}(\lambda) = b_{abs}(\lambda)_{liquid} + b_{abs}(\lambda)_{solid} \tag{4}$$

where $AAE_{liquid}$ and $AAE_{soild}$ are the AAE values of BC from liquid and solid
fuel combustion, $\lambda_1$ and $\lambda_2$ are of different wavelengths. The selection of
wavelengths (370-880 nm and 470-950 nm) can impact source apportionment
results. Here, the 470 nm and 950 nm were chosen as they were recommended
in the Aethalometer model (Drinovec et al., 2015). Moreover, source
apportionment result of the Aethalometer model highly depend on selection
of AAE pairs, with the value of AAE being determined by the type of biomass,
combustion processes and long-ranged transport condition (Gul et al., 2021).
The effect of different AAE values on the results discussed in section 3.3.2
(source diagnostic tracer). Combining the equations (2) ~ (4), we can obtain
the contribution of solid fuel combustion (BB%) to total BC:

$$BB(\%) = \frac{b_{abs}(\lambda_2)_{solid}}{b_{abs}(\lambda_2)} \times 100\% \tag{5}$$

Then, the $BC_{solid}$ can be obtained as follows:

$$BC_{solid} = BC(880nm) \times BB(\%) \tag{6}$$

Finally, the $BC_{liquid}$ can be calculated as:

$$BC_{liquid} = BC(880nm) - BC_{solid} \tag{7}$$

**2.3 Building random forest model and tuning hyper parameters**
The random forest (RF) machine learning algorithm is utilized to
reproduce historical time series data of BC. RF, a model comprising hundreds
of decision trees, splits data based on the informative features to avoid
overfitting, However, decision trees can easily overfit, resulting in inaccurate
model predictions. RF selects random samples of observation data for each
decision tree, a common problem in decision trees, by using random data





samples for each tree. The RF algorithm has been effectively applied in
atmospheric chemistry regions for predicting $PM_{10}$ and organic carbon (OC)
in different regions (Grange et al., 2018; Qin et al., 2022), demonstrating its
strong predictive capabilities.
In this work, the BC concentrations from 2019-2021 (target variables)
along with pollutants gases ($SO_2$, CO, $NO_2$) and meteorology factors such as
T, RH, WS, WD and BLH (independent variables) were inputted into RF
models. To train the RF model and assess the predicting ability of three RF
models, the whole dataset was randomly divided into training and testing sets
in a ratio of 8:2. Given that observational data followed a log-normal
distribution, most of the data are concentrated within a specific interval,
resulting a poor model performance on extreme values. To ensure a good
model performance, some data augmentation methods were used to achieve
data balance by interpolating or duplicating the less frequent data, ensuring
that the overall data essentially conforms to a uniform distribution (Hong et
al., 2023; Huang et al., 2023). To obtain optimal hyperparameter values, 10-
fold cross-validation was utilized on the training sets, dividing the datasets
into 10 subsamples, where 9 subsamples were used for training data and 1
subsample for testing. Optimized parameters for the models were chosen
based on the best mean squared error (MAE), root mean squared error (RMSE)
and R square ($R^2$) values obtained from the 10-fold cross-validation. Finally,
the test sets were then put into models and evaluated model predicting abilities.
The optimized parameters selected for the models are presented in Table 1.
The BC monitored by Aethalometer at 370 nm wavelength was also predicted
by RF models with same independent variables to explore the changes in BC
sources in Nanjing from 2014 to 2021.

Table 1 Parameters used in random forest models

| Parameters | Range | Optimal value | |
|---|---|---|---|
| | | BC_880nm | BC_370nm |
| n_estimators | [100-350] | 95 | 100 |
| max_depth | [10-30] | 25 | 23 |
| max_feature | [auto, sqrt, log2] | sqrt | sqrt |
| criterion | [friedman_mse, poisson, squared_error, absolute_error] | absolute_error | absolute_error |

**2.4 Kolmogorov-Zurbenko filter**



The KZ filter, a method for decomposing time series data into distinct
components, is widely utilized in air pollutants studies to differentiate the
influence of meteorology and emissions strength on the long-term trend of air
pollutants (Wise and Comrie, 2005; Yin et al., 2019; Chen et al., 2019). Since
the original concentration of BC follows a log-normal distribution, the data
($\chi$) were transformed into natural logarithmic form ($X = \ln (\chi)$) before
applying the KZ filter, allowing the data follow normally distribution (Zheng
et al., 2023). The KZ filter assumes that the original time series of a certain
air pollutant comprises short-term, seasonal, and long-term components. Thus,
the original time series of BC [X(t)] can be expressed as:

$$X(t) = E(t) + S(t) + W(t) \tag{8}$$

Here, E(t) represents the long-term component, mainly affected by climate,
long-range transport of air pollutants and emission intensity changes due to
shifts in energy structure. S(t) is the seasonal component, attributed to
variations in meteorology conditions and emission intensity across different
seasons. W(t) is the short-term component driven by weather patterns and
fluctuations in local-scale emissions.
The KZ filter is a low-pass filter characterized by a window length (m)
and iterations (p). Different 'm' and 'p' values can be used to separate each
component of an air pollutant. $KZ_{(15,5)}$ can eliminate cycles that are less than
33 days and obtain the baseline component of the original data. The W(t) can
be easily obtained by subtracting $X_{BL}(t)$ from X(t). Therefore, the long-term,
short-term and seasonal components can be extracted as follows:

$$X_{BL}(t) = KZ_{(15,5)}[X(t)] = X(t) - W(t) \tag{9}$$

The $X_{BL}$ is assumed to consist of its repeated climatological seasonal cycle
($X_{BL}^{clm}$) and residuals ($\varepsilon$).

$$X_{BL} = X_{BL}^{clm}(t) + \varepsilon \tag{10}$$

The $X_{BL}^{clm}$ contains most of the seasonality in $X_{BL}$, while $\varepsilon$ consist of E(t) along
with minor seasonal variability unconsidered in $X_{BL}^{clm}$. Applying a KZ filter
with a window length of 365 and an iteration of 3 ($KZ_{(365,3)}$) to $\varepsilon$, the E(t) and
S(t) can be obtained:

$$E(t) = KZ_{(365,3)}[\varepsilon(t)] = X_{BL}(t) - S(t) \tag{11}$$



Due to emissions and meteorological condition changes can be both influence
on long-term trend of BC, the long-term component can be assumed to consist
of emission-related ($E_{LT}^{emi}$) and meteorology-related ($E_{LT}^{met}$) components. Thus,
the $X_{BL}$ can be expressed as follows:

$$X_{BL}(t) = S(t) + E_{LT}^{emi} + E_{LT}^{met} \qquad (12)$$


To derive the $E_{LT}^{emi}$ in Eq.(9), the multiple linear regression model was
conducted baseline component of BC along with baseline components of six
meteorological factors such as T, RH, WS, WD, BLH, surface pressure (SP).
Then, the formulas can be written as follows:

$$X_{BL}(t) = a_0 + \sum_i a_i MET_{BL} + \varepsilon' \qquad (13)$$

Where $a_0$ is the intercepts of multiple linear regression model outcomes.
$MET_{BL}$ denote the baseline components of meteorology factors which are
obtained by KZ $_{(15,5)}$. $\varepsilon'$ is the sum of emission-related long-term variability
and some minor seasonal variability unexplained by the multiple linear
regression model. Therefore, the $E_{LT}^{emi}$ can be extracted by applying KZ $_{(365,3)}$
to $\varepsilon'$. Then, the $E_{LT}^{met}$ can be obtained by subtracting $E_{LT}^{emi}$ from long-term
component (E(t)) (Seo et al., 2018).

$$E_{LT}^{emi}(t) = KZ_{(365,3)}[\varepsilon'(t)] = E(t) - E_{LT}^{met}(t) \qquad (14)$$


**3 Results and Discussion**
**3.1 General characteristic of BC in Nanjing**
Figure 1(a) shows the hourly (dots) and daily (line) mean variation of
BC, $PM_{2.5}$ mass concentrations, and the proportion of BC to $PM_{2.5}$ in Nanjing.
A 400-fold variation was found in hourly BC concentration, which ranged
from 0.04 to 16.05 μg m$^{-3}$. Daily BC levels fluctuated much less than hourly
concentration, from the lowest value of 0.40 μg m$^{-3}$ (15$^{th}$ May 2021) to the
highest value of 9.58 μg m$^{-3}$ (24$^{th}$ January 2019). The average BC level during
the whole sampling period was 2.52 ± 1.62 μg m$^{-3}$. Figure 1(b) illustrates the
frequency distributions of hourly BC concentrations during different
sampling periods. Over three years, BC distributions shifted toward lower
values. In 2019, the most frequent BC concentrations were observed in 2 to 3
μg m$^{-3}$ range, accounting for 26.2% of samples. In 2020 and 2021, the most
BC levels were found in the 1 to 2 μg m$^{-3}$ range, with frequencies of 38.0%



and 41.9%, respectively. BC levels exceeding 7 μg m$^{-3}$ accounted for 5.1%, 0.8% and 0.01% in three years. PM$_{2.5}$ showed a similar variation to BC, with a significant correlation (r = 0.74, p < 0.05) observed between daily PM$_{2.5}$ and BC concentrations during sampling period. The hourly ratio of BC to PM$_{2.5}$ varied from 0.1 to 99%, with an annual mean of 12%. Compared to a previous study conducted in Yangtze River Delta, the BC/PM$_{2.5}$ ratio in Nanjing was much higher than Shanghai (5.6%) (Wei et al., 2020), implying a greater importance of primary emissions in Nanjing.

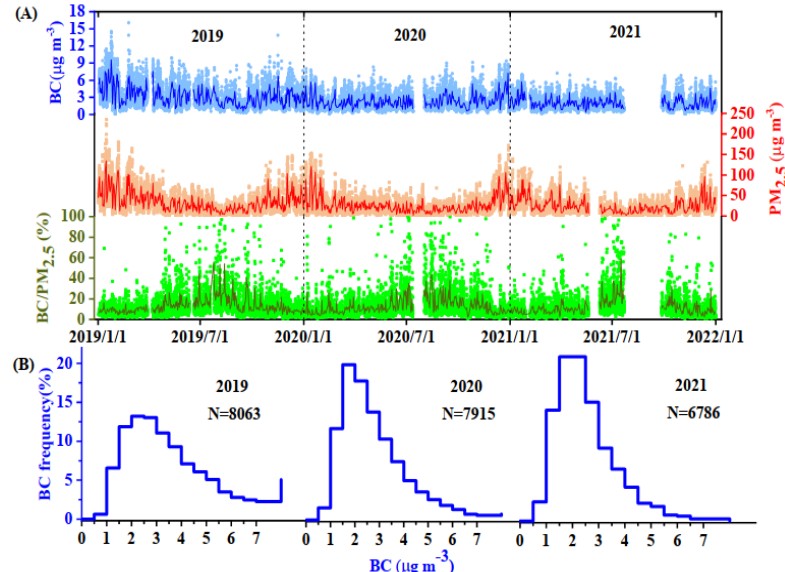

Figure 1 (A) Hourly (dots) and daily (line) concentration of BC, PM$_{2.5}$ and BC/PM$_{2.5}$ and (b) frequency of BC for each year during 2019, 2020 and 2021. N represents number of hourly BC concentration for one year

Table 2 listed long-time (equal or more than one year) BC mass concentrations monitored by optical method in Nanjing and other sampling sites all over the world from previous studies. Nanjing's three-year average BC level was the lowest among previous studies performed in Nanjing, indicating that primary emissions in Nanjing are decreasing year by year. While BC levels in other southern Chinese cities like Shanghai and Wuhan were at least 12.0% lower than those in Nanjing, they were at least 13.9% higher in northern Chinese cities like Beijing and Baoji. Additionally, BC concentrations in Nanjing were five times higher than in the baseline station



Mt. Waliguan.
Table 2 Comparison of BC mass concentration in Nanjing with other sites

| Location | Site type | Instrument | Study period (yyyy.mm) | BC (µg m$^{-3}$) | Reference |
|---|---|---|---|---|---|
| Nanjing, China | urban | AE33 | 2019.01-2021.12 | 2.52 ± 1.62 | Present study |
| Nanjing, China | suburban | AE31 | 2012.01-2012.12 | 4.2 ± 2.6 | (Zhuang et al., 2014) |
| Nanjing, China | urban | MAAP* | 2017.12-2018.11 | 2.8 ± 2.0 | (Zhang et al., 2020) |
| Mt. Waliguan, China | baseline | AE31 | 2008.01-2017.12 | 0.45 ± 0.37 | (Dai et al., 2021) |
| Beijing, China | urban | AE31 | 2016.01-2016.12 | 3.4 ± 3.0 | (Li et al., 2022) |
| Baoji, China | urban | AE31 | 2015.01-2015.12 | 2.9 ± 1.7 | (Zhou et al., 2018) |
| Xianghe, China | rural | AE31 | 2013.04-2015.03 | 5.4 ± 4.4 | (Ran et al., 2016) |
| Shanghai, China | urban | AE33 | 2017.01-2017.12 | 2.2 ± 1.3 | (Wei et al., 2020) |
| Wuhan, China | urban | AE33 | 2013.06-2018.12 | 1.4 ± 1.2 | (Zheng et al., 2020) |
| Panchgaon, India | suburban | AE42 | 2015.04-2016.03 | 7.2 ± 0.3 | (Dumka et al., 2019) |

*MAAP: Multi-angle absorption photometer

## 3.2 Temporal variation of BC mass concentrations in Nanjing

### 3.2.1 Interannual, seasonal, and monthly variations

The annual, seasonal, and monthly variations in BC mass concentrations
are illustrated in Figure 2. The annual average BC mass concentration in 2019
(3.2 ± 2.0 µg m$^{-3}$) was higher than in 2020 (2.3 ± 1.4 µg m$^{-3}$) and 2021 (2.0 ±
1.1 µg m$^{-3}$). A significant reduction of 28.1% in BC mass concentration was
observed from 2019 to 2020, much higher than the reduction (13.0%)
observed during 2020-2021. Similar, PM$_{2.5}$ concentrations reduced more
sharply during 2019-2020 (24.1%) than in 2020-2021 (6.2%). To prevent the



spread of COVID-19, a series of lockdown measures were imposed in China
in late January 2020, resulting in a remarkable decrease in concentrations of
air pollutants (Bauwens et al., 2020; Li et al., 2020; Wang et al., 2020).
Seasonally, the highest averaged BC level over three years occurred in
winter ($2.9 \pm 2.0$ μg m$^{-3}$), with no obvious variation identified in spring ($2.5$
$\pm 1.5$ μg m$^{-3}$), summer ($2.4 \pm 1.4$ μg m$^{-3}$) or autumn ($2.3 \pm 1.5$ μg m$^{-3}$),
suggesting a generally locally dominated source of BC emissions. The results
of bivariate polar plots showed the highest BC levels in low wind speed (WS
$< 4$ m s$^{-1}$) in all seasons (Figure S2), further indicating that local sources are
the predominant contributors to atmospheric BC in Nanjing. High BC mass
concentrations in winter are mainly caused by enhanced emissions due to cold
weather and deteriorating meteorological dispersion conditions resulting from
low temperatures. A similar seasonal pattern was also found in previous
studies conducted in other Yangtze River Delta cities like Shanghai and Hefei
(Chang et al., 2017; Zhang et al., 2015). Seasonal average concentrations of
BC varied from 1.83 (autumn of 2021) to 3.40 μg m$^{-3}$ (spring of 2019) across
different years. In 2019, BC concentration in spring ($3.4 \pm 1.9$ μg m$^{-3}$) was
higher than in winter ($2.6 \pm 1.5$ μg m$^{-3}$), likely due to decreased human
activities during the lockdown period. In contrast to the spring of 2019, higher
levels of BC were found in winter during 2020 and 2021.
The monthly mean concentrations of BC showed relatively large
variation, ranging from 1.6 (November of 2021) to 5.1 μg m$^{-3}$(January of
2019). The highest monthly average BC levels were found in January ($3.5 \pm$
$2.3$ μg m$^{-3}$), followed by December ($2.9 \pm 1.7$ μg m$^{-3}$). The monthly variation
pattern of BC is consistent with previous studies in Nanjing, which reported
the highest BC levels in January and December (Zhang et al., 2020; Xiao et
al., 2020). Additionally, the BC concentration in January was 37% higher than
in August ($2.2 \pm 1.1$ μg m$^{-3}$), attributed to relatively lower emission strength
and larger precipitation in summer in Nanjing.





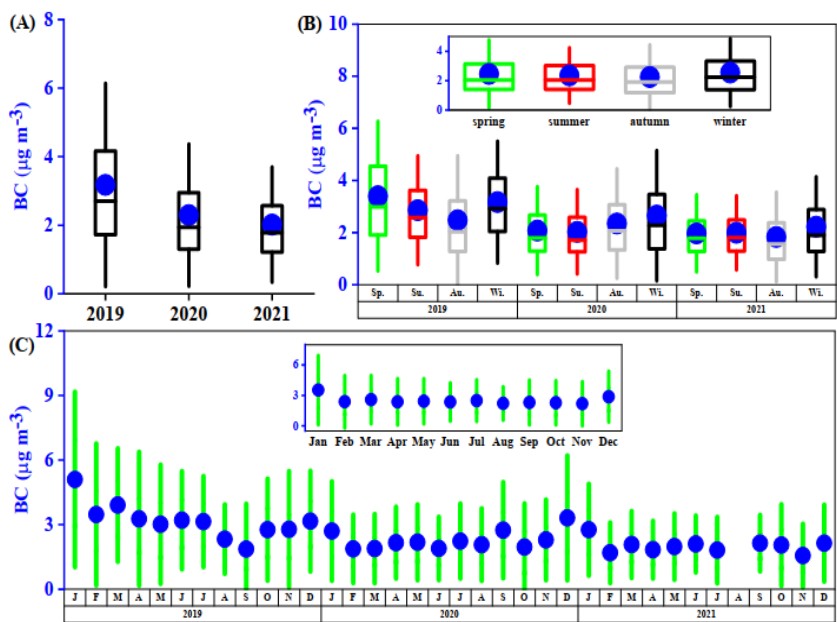

Figure 2 (A) Interannual, (B) seasonal, and (C) monthly variations of BC. The relatively small figures in (B) and (C) are overall average seasonal and monthly values. The blue dots represent average BC values. The rectangles in (A) and (B) represent the 25% and 75% quantiles. The vertical lines in (A), (B), and (C) represent 10% and 90% quantiles.

### 3.2.2 Diurnal variation of BC

The diurnal variations of BC mass concentrations for each year are plotted in Figure.3(a). The diurnal cycles of BC, like those in previous studies conducted in Nanjing (Xiao et al., 2020; Zhang et al., 2020; Zhuang et al., 2014), exhibited bimodal distributions in selected three years. BC mass concentrations remained relatively flat at midnight and then increased from 3:00 (local time, LT) to 7:00 LT. After reaching the highest values at 7:00 LT, BC levels decreased, reaching the lowest values at 16:00 LT, then increased again, and maintaining higher values in the evening. The bimodal diurnal patterns of BC were attributed to the intensity of emissions and variations in meteorological conditions (Cao et al., 2009). The morning peak of BC was mainly caused by vehicle emissions during the traffic rush hour, as indicated CO and NO$_2$ also showing similar diurnal cycles to BC (Figure S3). After the morning peak, the boundary layer height developed and WS increased, increasing atmospheric dilution capability and lowering the BC levels. After



14:00 LT, due to a decrease in boundary layer height and WS, BC was
gathered on the surface layer, resulting in higher BC loading from evening to
midnight. The peak BC concentration in 2019 was 29%, 38% higher than in
2020 and 2021 respectively, indicates air quality in Nanjing is getting better
due to the strict implementation of air pollution control plans. Additionally,
the impact of COVID-19 lockdown measures during selected years have also
contributed to the reduction in BC concentrations.
To further explore the impacts of human activities on ambient BC
concentrations, the diurnal variation in BC was separately investigated for
weekdays and weekends. As shown in Figure.3(b), the diurnal patterns of BC
on both weekdays and weekends exhibited bimodal distributions, with similar
peak times at morning vehicle rush hours (7:00 LT), suggesting that local
emission sources of BC in northern Nanjing do not differ significantly
between weekdays and weekends.

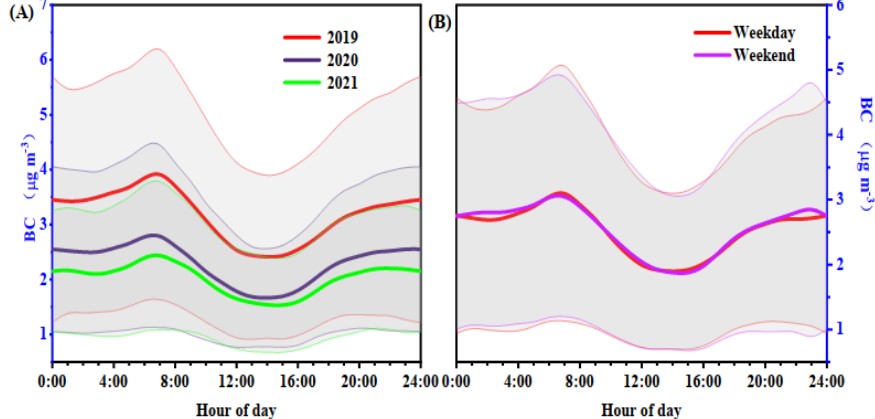


Figure 3 Diurnal variation of BC (A) for each year during 2019-2020 and (B) during weekdays and
weekends. Shaded areas represent the standard deviation at each time of day.
**3.3 Source apportionment of BC**
**3.3.1 Source apportionment of BC by Aethalometer model**
The AAE values, calculated by power-law fit between light absorbance
and seven wavelengths followed a lognormal distribution in selected three
years, with an hourly variation ranging from 0.71 to 2.59 (Figure S4). The
three-year average AAE value was $1.25 \pm 0.14$, with the highest value of 1.28
$\pm$ 0.13 in 2021, slightly higher by 4.0% and 4.3% in 2019 and 2020,





respectively, indicating similar BC emission sources during the sampling
period. Seasonally, the lowest AAE value of $1.13 \pm 0.14$ was found in summer,
while the highest AAE value of $1.32 \pm 0.11$ appeared in winter. The monthly
variation of AAE showed a valley in the summer months (particularly in July)
and high values in winter (December), suggesting that Nanjing was
predominantly influenced by traffic-related liquid fuel burning in summer,
and coal-related combustion in winter.
To quantify the relative contribution of liquid and solid fuel combustion
to BC concentration, the Aethalometer model, as mentioned in section 2.2,
was applied. The Aethalometer model was initially used for source apportion
BC in Europe, where fossil fuel and biomass burning emissions were two
major sources. However, China's energy structure differs from Europe's, with
coal combustion still playing a significant role. Liu et al. (2018) summarized
AAE values from different coal burning sources in China, finding that AAE
values of coal burning were close to those of biomass combustion. Thus, AAE
values of 1.0 for liquid fuel ($AAE_{liquid}$) and 2.0 for solid fuel ($AAE_{solid}$) were
selected for this work. The same AAE pairs were also used for source
apportionment of BC in previous study carried out in Nanjing (Lin et al.,
2021). Figure 4 shows the time series of absolute BC concentrations derived
from liquid and solid fuel combustion, along with a depiction of their relative
contributions to BC in different seasons for each year. The three-year average
concentration of $BC_{liquid}$ was $2.0 \pm 0.5 \mu g \ m^{-3}$, approximately four times that
of $BC_{solid}$. Liquid fuel combustion is the dominant source of BC in Nanjing,
with 79% of BC generated from the consumption of liquid fuel. Interannually,
the contributions of liquid fuel ranged from 76% to 81%, results that are
comparable to other cities in China such as Wuhan (81%) and Shanghai (88-
94%) (Zheng et al., 2020; Wei et al., 2020). The contribution of liquid fuel
burning to BC was highest in summer (85%), in contrast to the lowest
appeared in winter (72%), much higher than that of Beijing (35.7%) (Li et al.,
2022). Beijing is heavily affected by heating activities in winter, such as
power plants and residential heating using coal and biomass, resulting in
higher solid fuel emissions. The seasonal average contribution of BB varied
by 5%, from 19% to 24%, influenced by coal-fired emissions from
surrounding factories and long-range transport of domestic cooking emissions
in rural areas in the Yangtze River Delta region (Wei et al., 2020).



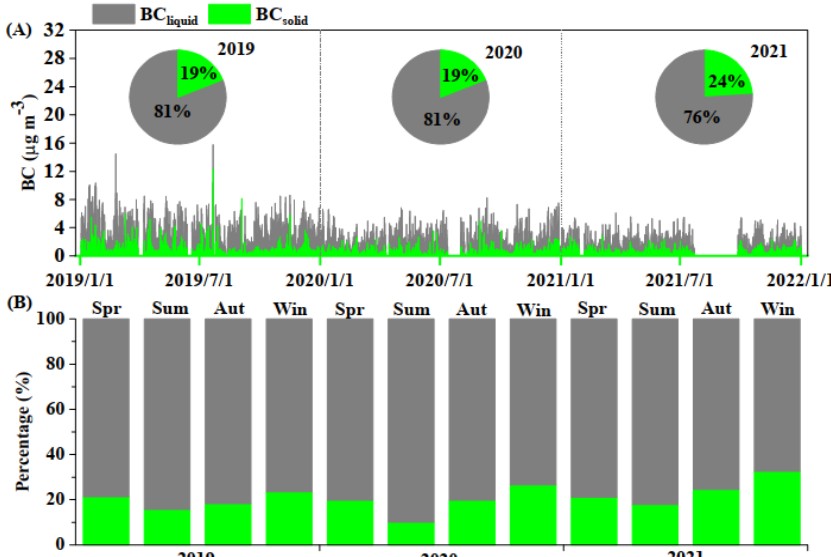

Figure 4 (A) Hourly variation of BC$_{liquid}$ and BC$_{solid}$, and (B) their relative contribution to BC. The pie charts in (A) are annual average relative contribution of BC$_{liquid}$ and BC$_{solid}$ to BC.

It is important to highlight that the results of the Aethalometer model are highly dependent on the determination of AAE values, with AAE$_{liquid}$ ranges between 0.8 to 1.1, and AAE$_{solid}$ values ranges between 1.8 to 2.2, as widely used in this model (Helin et al., 2018; Dumka et al., 2019; Fuller et al., 2014). To estimate the uncertainty of the Aethalometer model, we calculated source apportionment results using different AAE pairs, the results are showed in Table S1. The relative contributions of BC$_{liquid}$ and BC$_{solid}$ to BC ranged from 62% to 90% and 10% to 38%. Thus, an uncertainty estimation of 27.4% for the Aethalometer model results in this work. Although there are uncertainties in source apportionment results, our results indicate that liquid fuel combustion is the main source of BC in Nanjing during the study period.

**3.3.2 Source diagnostic tracers**

The ratios of BC/PM$_{2.5}$ and BC/CO (carbon monoxide, CO) have been utilized to estimate emission sources in previous studies since they can vary when emitted from different sources (Chow et al., 2011; Zhang et al., 2009). The proportion of BC in PM$_{2.5}$ is higher in traffic sources than that from other sources (such as residential coal combustion and forest fire). As listed in Table S2, higher BC/PM$_{2.5}$ ratios were found in heavy-duty diesel (33-74%) and



light-duty diesel (62-64%), followed by those from agricultural burning (6-13%) and forest fire (3%) (Table S2) (Chow et al., 2011). The highest ratio of $BC/PM_{2.5}$ appeared in autumn time (20%) while the lowest was observed in winter (8%), suggesting increased biomass and coal burning in winter. Previous studies reported that the ratio of BC/CO was lower in traffic emissions, as compared to the ratios from industry, power plant, residential and traffic emissions were 0.72%, 1.77%, 3.71%, and 0.52%, respectively (Table S2) (Zhang et al., 2009). The average ratios of BC/CO in spring, summer, autumn, and winter were 0.39%, 0.49%, 0.49%, and 0.31%, respectively, further suggesting that the traffic source was dominant in Nanjing (Table 3).

To further support the source apportionment results of BC, a correlation analysis was conducted between BC and trace gases such as $SO_2$, and $NO_2$, mainly derived from coal combustion, and vehicles emissions respectively. As listed in Table 3, the correlations of BC with $NO_2$ (0.54-0.67) were higher than the correlations of BC with $SO_2$ (0.16-0.59), further indicating the dominance of traffic emission in Nanjing.

Table 3 Mass ratios and correlations between BC and other pollutants

|  |  | Spring | Summer | Autumn | Winter | Annual |
|---|---|---|---|---|---|---|
| Mass ratios (%) | $BC/PM_{2.5}$ | 12.93 | 10.63 | 19.64 | 7.92 | 12.78 |
|  | BC/CO | 0.39 | 0.49 | 0.49 | 0.31 | 0.42 |
| Correlation | $BC-SO_2$ | 0.49 | 0.16 | 0.32 | 0.59 | 0.38 |
|  | $BC-NO_2$ | 0.66 | 0.61 | 0.54 | 0.67 | 0.60 |

## 3.4 Long-term trend of BC

### 3.4.1 Black carbon simulation results

After training the RF models with optimal hyperparameters, the models for BC at 880 nm and 370 nm were evaluated on test sets to assess predictive performance. The density scatter plot as displayed in Figure 5 showed that the RF model accurately reproduced hourly BC concentrations at both wavelengths. The RF model explained over 90% of the variation in BC concentrations, achieving an $R^2$ of 0.92 between the monitored and predicted results at both 370 and 880 nm. At the 370 nm wavelength, the RMSE was 0.57 μg m$^{-3}$, which was 22.8% higher than the RMSE at 880 nm, likely due to the higher observed BC levels at this wavelength. In addition to evaluating

the RF model using the test set, further validation was conducted using
Tracking Air Pollution in China (TAP) (10 km × 10 km, http://tapdata.org.cn)
data. The predicted BC values at 880 nm from the RF model showed good
agreement with the TAP dataset, with an R2 of 0.72 (Figure S5). Using the
trained model and available predictors, hourly BC concentration at the
sampling site can be accurately reconstructed for any given period, consistent
with AE33.

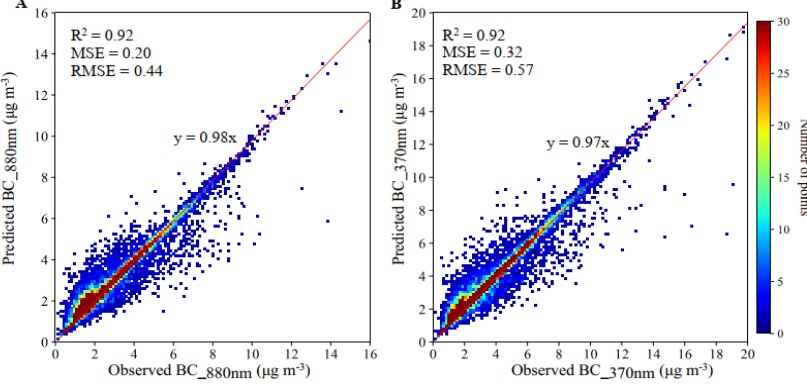

Figure 5 Density scatter plots of hourly observed and modeled BC at (a) 880 nm and (b) 370 nm
After training the RF models with input data, Shapley Additive
exPlanations (SHAP) values were used to assess the importance of each
predictors on model outcomes (Lundberg and Lee, 2017). Figure S6 presented
the ranked average SHAP values for each predictor for BC at the two
wavelengths. $NO_2$, BLH and $SO_2$ were identified as having the greatest
impact on model's prediction. Similar to BC, $NO_2$ and $SO_2$ are primarily
emitted from incomplete combustion processes involving fossil fuels (Lee et
al., 2017; Yao et al., 2002). As a result, BC, $NO_2$ and $SO_2$ are often co-emitted
by factories or traffic near the sampling site. BLH determines the diffusion
capacity of the atmosphere; a lower BLH means stronger atmospheric stability,
resulting in increased BC levels on the surface air. Unlike BLH, the
contribution of other meteorology predictors such as T, RH, WS and WD,
were relatively low compared to pollutant gases. One possible reason for this
is meteorological condition changes may not have an immediate effect on
atmospheric BC levels; instead, there may be a certain lag in their effects.

**3.4.2 Long-term temporal variation of BC**

Meteorological data and air pollutants concentrations were used in the





trained RF model to estimate BC concentrations at 370 and 880 nm from 2014
to 2021. The Aethalometer Model, using AAE of 1 and 2, was then applied to
the simulated BC to explore the long-term temporal variation of source-
specific BC. Between 2014 and 2021, average BC concentrations decreased
by 35.7% from $3.12 \pm 1.39$ µg m$^{-3}$ in 2014 to $2.04 \pm 0.33$ µg m$^{-3}$ in 2021. The
statistical significance of the reduction in BC and source-specific BC was
assessed using the Mann-Kendall test on monthly median values, with results
shown in Figure 6. A significant decreasing trend ($p<0.01$) in BC
concentrations was observed, with a slope of -0.13 µg m$^{-3}$yr$^{-1}$. Similar
reductions have also been reported across various regions in China since 2013
(He et al., 2023; Sun et al., 2022a; Chow et al., 2022; Dai et al., 2023).
Significantly decreases were also observed in BC$_{liquid}$ ($p<0.01$) and BC$_{solid}$
($p<0.05$) concentrations. From 2014 to 2021, BC$_{liquid}$, decreased by 38.4%
(from $2.55 \pm 1.14$ µg m$^{-3}$ to $1.57 \pm 0.89$ µg m$^{-3}$ in 2021) at an absolute rate of
-0.10 µg m$^{-3}$yr$^{-1}$, while BC$_{solid}$ decreased by 20.3% (from $0.59 \pm 0.52$ µg m$^{-3}$
to $0.47 \pm 0.33$ µg m$^{-3}$) at a rate of -0.03 µg m$^{-3}$yr$^{-1}$. The contributions of
different sources to the overall BC reduction were estimated by comparing
the absolute decrease slopes of BC$_{liquid}$ and BC$_{solid}$ to the overall BC decrease
slope. It was found that 77 % of total BC reduction was due to the decreased
liquid fuel combustion, highlighting the significant role of BC$_{liquid}$ in reducing
BC concentrations from 2014 to 2021.

525       Throughout the study period, BC concentrations exhibited two distinct
decrease trends, aligned with the implementation of the Air Pollution and
Control Action Plan (2013-2017, P1) and the Three-Year Action Plan (starting
in 2018, P2) by the Chinese government. To compare the decreasing trends of
BC in the two periods, the absolute trends were normalized by the average
values for each period. The change rates of BC and other air pollutants are
shown in Table 4. During P1, the relative slopes of BC and BC$_{liquid}$ were -
4.18 % yr$^{-1}$ ($p < 0.1$) and -4.26 % yr$^{-1}$ ($p < 0.05$), respectively BC$_{liquid}$
accounted for 83% of the total decrease in atmospheric BC concentrations.
Since the decrease in BC$_{solid}$ is not significant, the actual contribution of
BC$_{liquid}$ may be higher than estimated. Compared to P1, the decline in BC,
BC$_{liquid}$ and BC$_{solid}$ concentration during P2 was much steeper, reaching -10.9 %
yr$^{-1}$ ($p < 0.01$), -9.7 % yr$^{-1}$ ($p < 0.01$) and -11.1 % yr$^{-1}$ ($p < 0.1$), respectively.
In the S2 period, reductions in both BC$_{liquid}$ and BC$_{solid}$ contributed to the



overall decrease in BC concentration, with $BC_{liquid}$ still being the dominant factor, accounting for 71% of the total reduction. SO2 and NO2, which shared the same sources as BC, also decreased more rapidly in S2 (-31.6 % $yr^{-1}$ and -8.5 % $yr^{-1}$) compared to S1 (-9.3 % $yr^{-1}$ and -0.7 % $yr^{-1}$), suggesting that air pollutants have been decreasing much faster after 2018 than before.

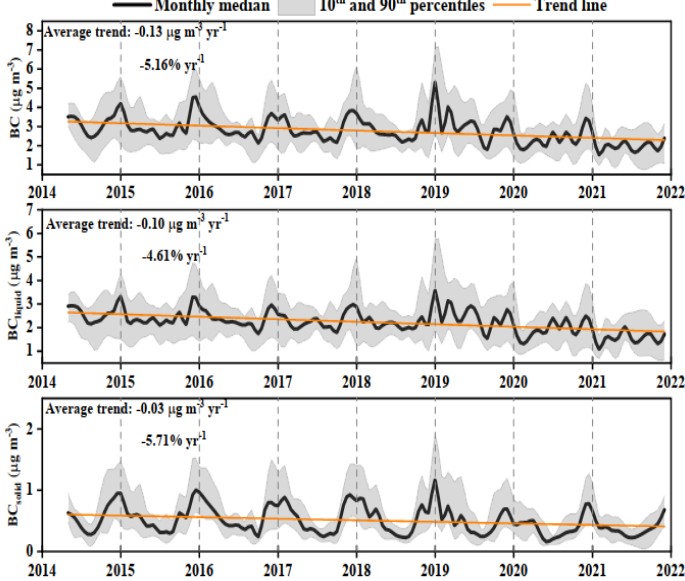

Figure 6 Trends in BC, $BC_{liquid}$ and $BC_{solid}$ at sampling site. The solid black line represents the monthly medians, the dash black lines represent the 10th and 90th monthly percentiles, and the orange line is the fitted long-term trend.

Table 4 The change rates of BC and other air pollutants during different period

| Study Period | air pollutants | absolute slope[a] | relative slope[b] | $p$ |
|---|---|---|---|---|
| | BC | -0.12 | -4.18 | 0.10 |
| | $BC_{liquid}$ | -0.10 | -4.26 | 0.05 |
| Air Pollution | $BC_{solid}$ | -0.02 | -3.47 | 0.60 |
| Prevention and | $PM_{2.5}$ | -11.90 | -25.28 | 0.01 |
| Control Action Plan | $NO_2$ | -0.29 | -0.73 | 0.90 |
| | $SO_2$ | -1.65 | -9.34 | 0.10 |
| | CO | 0.01 | 1.10 | 0.77 |
| | BC | -0.28 | -10.85 | 0.01 |
| | $BC_{liquid}$ | -0.20 | -9.71 | 0.01 |
| After 2018 | $BC_{solid}$ | -0.05 | -11.06 | 0.10 |
| | $PM_{2.5}$ | -4.33 | -14.96 | 0.05 |
| | $NO_2$ | -3.06 | -8.49 | 0.05 |



| | | | |
|---|---|---|---|
| SO$_2$ | -2.36 | -31.58 | 0.01 |
| CO | 0.02 | 2.73 | 0.64 |

[a]: µg m$^{-3}$ yr$^{-1}$
[b]: % yr-1
The seasonal trends in BC and its different sources were further
investigated in Nanjing. As shown in Figure 7, significant reductions in BC
concentrations were observed across all seasons. The decreasing slopes of BC
in spring (-6.1 % yr$^{-1}$, $p < 0.05$) and winter (-6.4 % yr$^{-1}$, $p < 0.01$) were steeper
than those in summer (-3.1 % yr$^{-1}$, $p < 0.1$) and autumn (-3.9 % yr$^{-1}$, $p < 0.01$).
The reduction rate of PM$_{2.5}$ in spring (-18.9 % yr-1, $p < 0.05$), summer (-
22.5%, $p < 0.05$) and autumn (-15.9 % yr-1, $p < 0.1$) was3 to 6 times that of
BC (Table S3). In winter, the reduction rate (-9.8 % yr-1, $p < 0.01$) is closer
to that of BC, suggesting that the reduction of primary pollutants in Nanjing
during winter might be more effective compared to other seasons. The
seasonal variation of BC$_{liquid}$ showed distinct trends across different seasons.
Significant reductions were observed in spring, autumn and winter, with the
absolute slope of -5.9 % yr$^{-1}$ ($p < 0.1$), -3.8 % yr$^{-1}$ ($p < 0.05$) and -6.5 % yr$^{-1}$
($p < 0.05$), respectively. BC$_{liquid}$ in summer was not statistically significant,
which may be partly due to increased traffic activity during tourism peak
season, leading to higher liquid fuel consumption. Moreover, the reduction
rate of PM$_{2.5}$ was faster in summer compared to BC, indicating that secondary
aerosol reductions were more pronounced during this season. BC$_{solid}$ showed
a similar decreasing slope in summer (-6.1 % yr$^{-1}$, $p < 0.01$), and winter (-6.2 %
yr$^{-1}$, p< 0.05), while autumn appeared relatively slower reduction (-4.2 % yr$^{-}$
$^1$). Similar to BC$_{solid}$, SO$_2$ exhibited a steeper change rate in winter (-24.4 %
yr$^{-1}$, $p < 0.01$) and a slower change rate in autumn (-16.2 % yr$^{-1}$, $p < 0.01$)
(Table S3). The reduction of BC$_{solid}$ in spring was not significant, which may
be influenced by long-range transport of biomass burning, as well as increased
agricultural activities during this season. It is worth noting that BC$_{liquid}$
contributed 76% to overall BC reduction in spring and BC$_{solid}$ contributed 25%
to overall BC reduction in summer. However, since the decreasing trend of
BC$_{solid}$ in spring and BC$_{liquid}$ in summer were not statistically significant, these
contributions may have been underestimated.





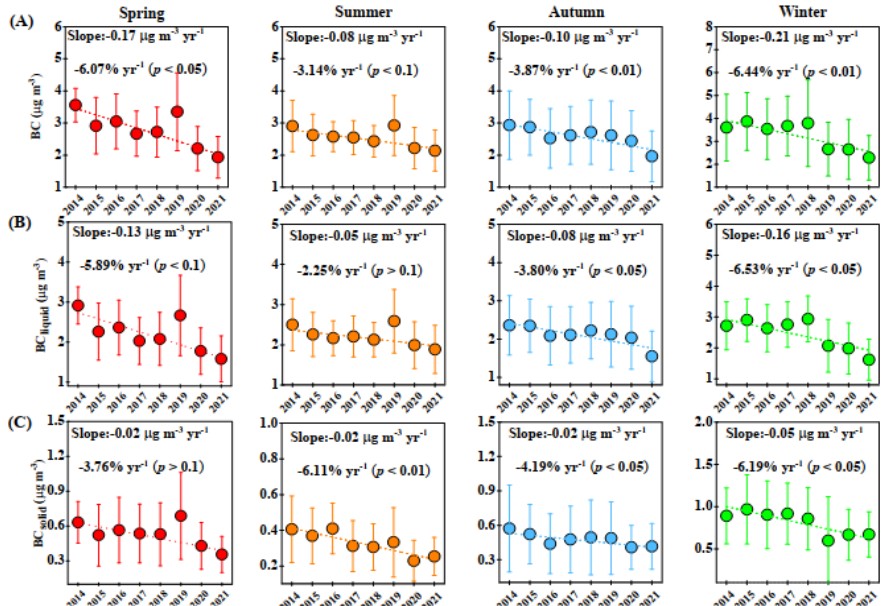

Figure 7 Seasonal variation of (A) BC, (B) $BC_{liquid}$ and (C) $BC_{solid}$ in spring, summer, autumn and winter. The circle in different color represents the average concentration of BC, $BC_{liquid}$ and $BC_{solid}$. The vertical lines represent the standard deviations of BC, $BC_{liquid}$ and $BC_{solid}$.

### 3.4.3 The impact of Emission and Meteorology

In addition to changes in emissions, meteorological conditions can also affect the long-term trends of pollutants by influencing their long-range transport and processes of dry and wet deposition. To explore these impacts on the long-term trends of BC, the KZ filter was applied to distinguish between emission-related ($E_{LT}^{emi}$) and meteorology-related ($E_{LT}^{met}$) trends. The daily averaged log-transformed original time series along with decoupled short-term, baseline and seasonal of BC were described in Figure S7. The short-term component of BC displayed notable fluctuations, while the seasonal component showed a clear cycle with higher levels in winter and lower levels in summer. The largest variances for BC (69%), $BC_{liquid}$ (73%) and $BC_{solid}$ (52%) were found in short-term component, reflecting the essential role of synoptic weather on the daily variations of primary aerosol content in Nanjing (Table S4). $BC_{solid}$ exhibits seasonal dependence with relatively higher seasonal component (40%) than BC (16%) and $BC_{liquid}$ (12%). The sum of variances explained by the short-term, seasonal and long-



term component for BC, $BC_{liquid}$ and $BC_{solid}$ were 93%, 92% and 92%, respectively. A total variance close to 100% indicating that these three components are largely independent of each other, suggesting that most of the meteorological influence have been effectively accounted and removed (Chen et al., 2019; Sun et al., 2022b; Zheng et al., 2020). To separate emission-related ($E_{LT}^{emi}$) and meteorology-related components ($E_{LT}^{met}$) from the long-term component ($E_{LT}$), multiple linear regression was conducted using the baseline component of meteorological parameters and BC. The model incorporating these meteorological parameters accurately reproduced the baseline of $BC_{solid}$ ($R^2 = 0.84$, $p < 0.001$). In contrast, it was less effective in explaining the baseline for BC ($R^2 = 0.59$, $p < 0.001$) and $BC_{liquid}$ ($R^2 = 0.51$, $p < 0.001$), suggesting that local emission changes across different seasons play an important role in impacting BC and $BC_{liquid}$ in Nanjing.

The linear trends of $E_{LT}$, $E_{LT}^{emi}$ and $E_{LT}^{met}$ for BC, $BC_{liquid}$ and $BC_{solid}$ are summarized in Table 5. It is important to note that the linear trend slope of $E_{LT}$ represents relative change rate (% $yr^{-1}$) of the baseline concentration, since original time series of BC were log-transformed before applying the KZ filter. To concert the fractional change rate into an absolute change rate (μg $m^{-3}$ $yr^{-1}$), it is multiplied by the average baseline concentration (not log-transformed). The $E_{LT}$ of BC and its distinct source exhibited significant ($p < 0.01$) declining trends, with slopes of -0.1, -0.08 and -0.02 ug $m^{-3}$ $yr^{-1}$ for BC, $BC_{liquid}$ and $BC_{solid}$, respectively. $BC_{liquid}$ was the dominant contributor to BC reduction, accounting for 80% of the overall decrease, suggesting that when the influence of seasonal and synoptic variations is excluded, its contribution to BC temporal variations becomes more evident. In addition, the relative contributions of $E_{LT}^{emi}$ and $E_{LT}^{met}$ to BC reduction were quantified by calculating the ratio of their absolute slopes to that of $E_{LT}$ (Zheng et al., 2023). Both meteorology conditions and emission reductions played crucial roles in reducing BC and its specific sources. Emission reductions were found to be the major contributor to the decline in long-term trends of BC, $BC_{liquid}$ and $BC_{solid}$, with contributions of 70%, 63% and 86%, respectively. While emissions reductions dominated the decrease in BC concentrations throughout the study period, their relative influence compared to meteorological conditions varied between the P1 (before 2018) and P2 (after 2018) phases. As shown in Figure 8, emission reductions played a more




prominent role, contributing 78%, 62% and 86% to the reductions in BC,
BC_{liquid} and BC_{solid}, respectively. However, during P2, meteorological
conditions played a leading role in reducing BC and BC_{liquid}, contributing 66%
and 70%, respectively. Moreover, meteorology condition had a notable
impact on BC_{solid} in P2, with its contribution increasing from 14% in P1 to
31%. This suggests that the rapid reduction of BC in P2 was largely due to
favorable meteorological conditions, which played a crucial role in
facilitating its decline.
Table 5 Linear trends of long-term component of BC and its sources including BC_{liquid} and BC_{solid}

| Componants | BC | | | BC_{liquid} | | | BC_{solid} | | |
|---|---|---|---|---|---|---|---|---|---|
| | absolute[a] | relative[b] | $p$ | absolute[a] | relative[b] | $p$ | absolute[a] | relative[b] | $p$ |
| $E_{LT}$ | -0.10 | -3.76 | 0.01 | -0.08 | -3.54 | 0.01 | -0.014 | -4.91 | 0.01 |
| $E_{LT}^{EMI}$ | -0.07 | -2.63 | 0.01 | -0.05 | -2.20 | 0.01 | -0.012 | -3.62 | 0.01 |
| $E_{LT}^{MET}$ | -0.03 | -1.13 | 0.01 | -0.03 | -1.32 | 0.01 | -0.002 | -1.27 | 0.01 |

[a]: $\mu g\ m^{-3}\ yr^{-1}$
[b]: % yr-1

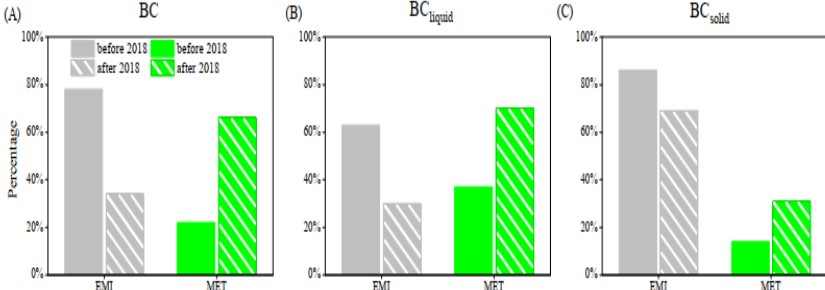

Figure 8 Contributions of Emission Reduction Policies and Meteorological Conditions to the
Decrease in BC Concentrations Before and After 2018. The (A), (B) and (C) panels represent BC,

649                                BC_{liquid} and BC_{solid}.

**4. Conclusion**
In this work, BC mass concentrations were continuously monitored in
Nanjing, China, from 2019 to 2021. Combining observations with random
forest algorithm, the BC concentrations from 2014-2021were reconstructed
to explore the long-term trends of BC and its sources during two distinct
emission reduction periods. The results showed that BC concentrations were
analyzed to reveal its characteristics and sources. The annual average BC
mass concentration during the study period was 2.5 ± 1.6μg m^{-3}. Relatively



higher BC mass concentrations were found in winter, while no clear variation was observed during other seasons, implying a locally dominant BC source. Diurnal variations showed a bimodal pattern with lower concentrations in daytime and higher values in night, primarily influenced by traffic rush hours and boundary layer heights. Liquid fuel combustion contributed more than 75% to BC in all years, with the highest contribution appearing in summer (85%) and the lowest in winter (72%).

The RF models explained over 90% variation and accurately captured seasonal cycle well of BC at 880 nm, demonstrating the strong predictive capability of the trained models. The long-term trend of BC, $BC_{liquid}$ and $BC_{solid}$ all exhibited significant ($p < 0.05$) declines, with $BC_{liquid}$ contributing the most to the overall BC reduction, accounting for 77% of the total decrease over entire period. Notably, BC levels decreased most rapidly during winter, while the reduction in summer was much slower. The trend in BC reduction varied between two distinct phases, in P2 (after 2018), BC levels declined much steeper compared to that in P1 (2014-2017), indicating that policies aimed at replacing coal to cleaner energy have been particularly effective in reducing primary pollutants. Over the entire period, emission reduction was the primary driver of BC reduction, contributing to BC, $BC_{liquid}$ and $BC_{solid}$ reduction, with contribution of 70%, 63%, and 86%, respevtively while meteorological conditions accounted for 30%, 37% and 24%. Although emission reduction dominated BC reduction over the entire period, the contributions of emission reduction and meteorological conditions to BC reduction differed between the two phases. In P1, emission reduction played a dominant role, while in P2, meteorological conditions became the primary driver of BC reduction. Our results highlight that to further reduce atmospheric BC, targeted policies should be implemented to restrict liquid fuel combustion, especially during the summer. Additionally, the impact of meteorological factors on BC concentrations should not be overlooked during emission reduction efforts.

**Data Availability**

The hourly meteorological reanalysis data ERA5 are available in the ECMWF at https://cds.climate.copernicus.eu/cdsapp#!/dataset/reanalysis-era5-single-levels?tab=form. Hourly averaged concentrations of $PM_{2.5}$, CO, $SO_2$ and $NO_2$ were obtained from https://quotsoft.net/air/. All the observational and



predicted data were openly accessible at the Open Science Framework
https://osf.io/8n32t/.

**Competing interests**

The contact author has declared that none of the authors has any competing
interests.

**Author contributions**

Yanlin Zhang designed the research. Fang Cao, Mingyuan Yu, and Chaman
Gul took part in data analysis and revised and commented on the paper.
Abudurexiati·Abulimiti wrote the paper. Yihang Hong analysis the data. All
authors contributed to the discussion of this paper.

**Acknowledgement**

This research was financially supported by the National Natural Science
Foundation of China (No. 42192512, 42107123, 42273087).

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

Contributions to Atmospheric Nitrate in Urban China from Observation to Prediction,
Environmental Science & Technology, 57, 18172-18182, 10.1021/acs.est.3c01651, 2023.
Fuller, G. W., Tremper, A. H., Baker, T. D., Yttri, K. E., and Butterfield, D.: Contribution of wood
burning       to       PM10       in       London,       Atmospheric       Environment,       87,       87-94,
https://doi.org/10.1016/j.atmosenv.2013.12.037, 2014.
Grange, S. K., Carslaw, D. C., Lewis, A. C., Boleti, E., and Hueglin, C.: Random forest meteorological
normalisation models for Swiss PM10 trend analysis, Atmos. Chem. Phys., 18, 6223-6239,
10.5194/acp-18-6223-2018, 2018.
Gul, C., Mahapatra, P. S., Kang, S., Singh, P. K., Wu, X., He, C., Kumar, R., Rai, M., Xu, Y., and Puppala,
S. P.: Black carbon concentration in the central Himalayas: Impact on glacier melt and potential



source contribution, Environmental Pollution, 275, 116544,
https://doi.org/10.1016/j.envpol.2021.116544, 2021.
He, C., Niu, X., Ye, Z., Wu, Q., Liu, L., Zhao, Y., Ni, J., Li, B., and Jin, J.: Black carbon pollution in China
from 2001 to 2019: Patterns, trends, and drivers, Environmental Pollution, 324, 121381,
https://doi.org/10.1016/j.envpol.2023.121381, 2023.
Helin, A., Niemi, J. V., Virkkula, A., Pirjola, L., Teinilä, K., Backman, J., Aurela, M., Saarikoski, S., Rönkkö,
T., Asmi, E., and Timonen, H.: Characteristics and source apportionment of black carbon in the
Helsinki metropolitan area, Finland, Atmospheric Environment, 190, 87-98,
https://doi.org/10.1016/j.atmosenv.2018.07.022, 2018.
Hong, Y., Zhang, Y., Bao, M., Fan, M., Lin, Y. C., Xu, R., Shu, Z., Wu, J. Y., Cao, F., Jiang, H., Cheng,
Z., Li, J., and Zhang, G.: Nitrogen-Containing Functional Groups Dominate the Molecular
Absorption of Water-Soluble Humic-Like Substances in Air From Nanjing, China Revealed by the
Machine Learning Combined FT-ICR-MS Technique, Journal of Geophysical Research:
Atmospheres, 128, 10.1029/2023JD039459, 2023.
Huang, Z.-J., Li, H., Luo, J.-Y., Li, S., and Liu, F.: Few-Shot Learning-Based, Long-Term Stable,
Sensitive Chemosensor for On-Site Colorimetric Detection of Cr(VI), Analytical Chemistry, 95,
6156-6162, 10.1021/acs.analchem.3c00604, 2023.
IPCC: Climate Change 2022 – Impacts, Adaptation and Vulnerability: Working Group II
Contribution to the Sixth Assessment Report of the Intergovernmental Panel on Climate Change,
Cambridge University Press, Cambridge, 10.1017/9781009325844, 2023.
Jiang, X., Li, G., and Fu, W.: Government environmental governance, structural adjustment and air
quality: A quasi-natural experiment based on the Three-year Action Plan to Win the Blue Sky
Defense War, Journal of Environmental Management, 277, 111470,
https://doi.org/10.1016/j.jenvman.2020.111470, 2021.
Lee, B. P., Louie, P. K. K., Luk, C., and Chan, C. K.: Evaluation of traffic exhaust contributions to
ambient carbonaceous submicron particulate matter in an urban roadside environment in Hong
Kong, Atmos. Chem. Phys., 17, 15121-15135, 10.5194/acp-17-15121-2017, 2017.
Li, L., Li, Q., Huang, L., Wang, Q., Zhu, A., Xu, J., Liu, Z., Li, H., Shi, L., Li, R., Azari, M., Wang, Y.,
Zhang, X., Liu, Z., Zhu, Y., Zhang, K., Xue, S., Ooi, M. C. G., Zhang, D., and Chan, A.: Air quality
changes during the COVID-19 lockdown over the Yangtze River Delta Region: An insight into the
impact of human activity pattern changes on air pollution variation, Science of The Total
Environment, 732, 139282, https://doi.org/10.1016/j.scitotenv.2020.139282, 2020.
Li, R., Han, Y., Wang, L., Shang, Y., and Chen, Y.: Differences in oxidative potential of black carbon
from three combustion emission sources in China, Journal of Environmental Management, 240,
57-65, https://doi.org/10.1016/j.jenvman.2019.03.070, 2019.
Li, W., Liu, X., Duan, F., Qu, Y., and An, J.: A one-year study on black carbon in urban Beijing:
Concentrations, sources and implications on visibility, Atmospheric Pollution Research, 13, 101307,
https://doi.org/10.1016/j.apr.2021.101307, 2022.
Li, Y., Lei, L., Sun, J., Gao, Y., Wang, P., Wang, S., Zhang, Z., Du, A., Li, Z., Wang, Z., Kim, J. Y., Kim,
H., Zhang, H., and Sun, Y.: Significant Reductions in Secondary Aerosols after the Three-Year
Action Plan in Beijing Summer, Environmental Science & Technology, 57, 15945-15955,
10.1021/acs.est.3c02417, 2023.
Lin, Y.-C., Zhang, Y.-L., Xie, F., Fan, M.-Y., and Liu, X.: Substantial decreases of light absorption,



concentrations and relative contributions of fossil fuel to light‑absorbing carbonaceous aerosols
attributed to the COVID-19 lockdown in east China, Environmental Pollution, 275, 116615,
https://doi.org/10.1016/j.envpol.2021.116615, 2021.
Liu, D., He, C., Schwarz, J. P., and Wang, X.: Lifecycle of light‑absorbing carbonaceous aerosols in
the atmosphere, npj Climate and Atmospheric Science, 3, 40, 10.1038/s41612-020-00145-8, 2020.
Liu, Y., Yan, C., and Zheng, M.: Source apportionment of black carbon during winter in Beijing,
Science of The Total Environment, 618, 531‑541, https://doi.org/10.1016/j.scitotenv.2017.11.053,
826 2018.

Lundberg, S. M. and Lee, S.-I.: A unified approach to interpreting model predictions, Advances in
neural information processing systems, 30, 2017.
Qin, Y., Ye, J., Ohno, P., Liu, P., Wang, J., Fu, P., Zhou, L., Li, Y. J., Martin, S. T., and Chan, C. K.:
Assessing the Nonlinear Effect of Atmospheric Variables on Primary and Oxygenated Organic
Aerosol Concentration Using Machine Learning, ACS Earth and Space Chemistry, 6, 1059‑1066,
10.1021/acsearthspacechem.1c00443, 2022.
Ramanathan, V. and Carmichael, G.: Global and regional climate changes due to black carbon,
Nature Geoscience, 1, 221‑227, 10.1038/ngeo156, 2008.
Ran, L., Deng, Z. Z., Wang, P. C., and Xia, X. A.: Black carbon and wavelength‑dependent aerosol
absorption in the North China Plain based on two‑year aethalometer measurements, Atmospheric
Environment, 142, 132‑144, https://doi.org/10.1016/j.atmosenv.2016.07.014, 2016.
Sandradewi, J., Prévôt, A. S. H., Szidat, S., Perron, N., Alfarra, M. R., Lanz, V. A., Weingartner, E., and
Baltensperger, U.: Using Aerosol Light Absorption Measurements for the Quantitative
Determination of Wood Burning and Traffic Emission Contributions to Particulate Matter,
Environmental Science & Technology, 42, 3316‑3323, 10.1021/es702253m, 2008.
Sarigiannis, D., Karakitsios, S. P., Zikopoulos, D., Nikolaki, S., and Kermenidou, M.: Lung cancer risk
from PAHs emitted from biomass combustion, Environ Res, 137, 147‑156,
10.1016/j.envres.2014.12.009, 2015.
Seo, J., Park, D. S. R., Kim, J. Y., Youn, D., Lim, Y. B., and Kim, Y.: Effects of meteorology and emissions
on urban air quality: a quantitative statistical approach to long‑term records (1999–2016) in Seoul,
South Korea, Atmos. Chem. Phys., 18, 16121‑16137, 10.5194/acp-18-16121-2018, 2018.
Sun, J., Wang, Z., Zhou, W., Xie, C., Wu, C., Chen, C., Han, T., Wang, Q., Li, Z., Li, J., Fu, P., Wang, Z.,
and Sun, Y.: Measurement report: Long‑term changes in black carbon and aerosol optical
properties from 2012 to 2020 in Beijing, China, Atmos. Chem. Phys., 22, 561‑575, 10.5194/acp-
22-561-2022, 2022a.
Sun, X., Zhao, T., Bai, Y., Kong, S., Zheng, H., Hu, W., Ma, X., and Xiong, J.: Meteorology impact on
PM2.5 change over a receptor region in the regional transport of air pollutants: observational
study of recent emission reductions in central China, Atmos. Chem. Phys., 22, 3579‑3593,
10.5194/acp-22-3579-2022, 2022b.
Wang, Y., Yuan, Y., Wang, Q., Liu, C., Zhi, Q., and Cao, J.: Changes in air quality related to the
control of coronavirus in China: Implications for traffic and industrial emissions, Science of The
Total Environment, 731, 139133, https://doi.org/10.1016/j.scitotenv.2020.139133, 2020.
Wei, C., Wang, M. H., Fu, Q. Y., Dai, C., Huang, R., and Bao, Q.: Temporal Characteristics and
Potential Sources of Black Carbon in Megacity Shanghai, China, Journal of Geophysical Research:
Atmospheres, 125, e2019JD031827, https://doi.org/10.1029/2019JD031827, 2020.



Wise, E. K. and Comrie, A. C.: Extending the Kolmogorov–Zurbenko Filter: Application to Ozone,
Particulate Matter, and Meteorological Trends, Journal of the Air & Waste Management
Association, 55, 1208-1216, 10.1080/10473289.2005.10464718, 2005.
Wu, B., Wu, C., Ye, Y., Pei, C., Deng, T., Li, Y. J., Lu, X., Wang, L., Hu, B., Li, M., and Wu, D.: Long-
term hourly air quality data bridging of neighboring sites using automated machine learning: A
case study in the Greater Bay area of China, Atmospheric Environment, 321, 120347,
https://doi.org/10.1016/j.atmosenv.2024.120347, 2024.
Xiao, S., Yu, X., Zhu, B., Kumar, K. R., Li, M., and Li, L.: Characterization and source apportionment
of black carbon aerosol in the Nanjing Jiangbei New Area based on two years of measurements
from        Aethalometer,       Journal      of      Aerosol       Science,       139,      105461,
https://doi.org/10.1016/j.jaerosci.2019.105461, 2020.
Yao, X., Chan, C. K., Fang, M., Cadle, S., Chan, T., Mulawa, P., He, K., and Ye, B.: The water-soluble
ionic composition of PM2.5 in Shanghai and Beijing, China, Atmospheric Environment, 36, 4223-
4234, https://doi.org/10.1016/S1352-2310(02)00342-4, 2002.
Yin, C., Deng, X., Zou, Y., Solmon, F., Li, F., and Deng, T.: Trend analysis of surface ozone at
suburban     Guangzhou,      China,     Science     of     The     Total     Environment,     695,     133880,
https://doi.org/10.1016/j.scitotenv.2019.133880, 2019.
Yu, M., Zhang, Y.-L., Xie, T., Song, W., Lin, Y.-C., Zhang, Y., Cao, F., Yang, C., and Szidat, S.:
Quantification of fossil and non-fossil sources to the reduction of carbonaceous aerosols in the
Yangtze River Delta, China: Insights from radiocarbon analysis during 2014–2019, Atmospheric
Environment, 292, 119421, https://doi.org/10.1016/j.atmosenv.2022.119421, 2023.
Zhang, L., Shen, F., Gao, J., Cui, S., Yue, H., Wang, J., Chen, M., and Ge, X.: Characteristics and
potential sources of black carbon particles in suburban Nanjing, China, Atmospheric Pollution
Research, 11, 981-991, https://doi.org/10.1016/j.apr.2020.02.011, 2020.
Zhang, Q., Streets, D. G., Carmichael, G. R., He, K. B., Huo, H., Kannari, A., Klimont, Z., Park, I. S.,
Reddy, S., Fu, J. S., Chen, D., Duan, L., Lei, Y., Wang, L. T., and Yao, Z. L.: Asian emissions in 2006
for the NASA INTEX-B mission, Atmos. Chem. Phys., 9, 5131-5153, 10.5194/acp-9-5131-2009,
889  2009.
Zhang, Q., Zheng, Y., Tong, D., Shao, M., Wang, S., Zhang, Y., Xu, X., Wang, J., He, H., Liu, W., Ding,
Y., Lei, Y., Li, J., Wang, Z., Zhang, X., Wang, Y., Cheng, J., Liu, Y., Shi, Q., Yan, L., Geng, G., Hong, C.,
Li, M., Liu, F., Zheng, B., Cao, J., Ding, A., Gao, J., Fu, Q., Huo, J., Liu, B., Liu, Z., Yang, F., He, K., and
Hao, J.: Drivers of improved PM2.5 air quality in China from 2013 to 2017,
Proceedings    of    the    National    Academy    of    Sciences,    116,    24463-24469,
doi:10.1073/pnas.1907956116, 2019.
Zhang, X., Rao, R., Huang, Y., Mao, M., Berg, M. J., and Sun, W.: Black carbon aerosols in urban
central China, Journal of Quantitative Spectroscopy and Radiative Transfer, 150, 3-11,
https://doi.org/10.1016/j.jqsrt.2014.03.006, 2015.
Zhang, Y.-L., Li, J., Zhang, G., Zotter, P., Huang, R.-J., Tang, J.-H., Wacker, L., Prévôt, A. S. H., and
Szidat, S.: Radiocarbon-Based Source Apportionment of Carbonaceous Aerosols at a Regional
Background Site on Hainan Island, South China, Environmental Science & Technology, 48, 2651-
2659, 10.1021/es4050852, 2014.
Zhao, C., Wang, Q., Ban, J., Liu, Z., Zhang, Y., Ma, R., Li, S., and Li, T.: Estimating the daily PM2.5
concentration in the Beijing-Tianjin-Hebei region using a random forest model with a 0.01° × 0.01°



spatial       resolution,       Environment       International,       134,       105297,
https://doi.org/10.1016/j.envint.2019.105297, 2020.
Zheng, H., Kong, S., Zhai, S., Sun, X., Cheng, Y., Yao, L., Song, C., Zheng, Z., Shi, Z., and Harrison, R.
M.: An intercomparison of weather normalization of PM2.5 concentration using traditional
statistical methods, machine learning, and chemistry transport models, npj Climate and
Atmospheric Science, 6, 214, 10.1038/s41612‑023‑00536‑7, 2023.
Zheng, H., Kong, S. F., Zheng, M. M., Yan, Y., Yao, L., Zheng, S., Yan, Q., Wu, J., Cheng, Y., Chen, N.,
Bai, Y., Zhao, T., Liu, D., Zhao, D., and Qi, S.: A 5.5‑year observations of black carbon aerosol at a
megacity in Central China: Levels, sources, and variation trends, Atmospheric Environment, 232,
117581, https://doi.org/10.1016/j.atmosenv.2020.117581, 2020.
Zhou, B., Wang, Q., Zhou, Q., Zhang, Z., Wang, G., Fang, N., Li, M., and Cao, J.: Seasonal
Characteristics of Black Carbon Aerosol and its Potential Source Regions in Baoji, China, Aerosol
and Air Quality Research, 18, 397‑406, 10.4209/aaqr.2017.02.0070, 2018.
Zhou, Y., Ma, X., Tian, R., and Wang, K.: Seasonal transition of Black carbon aerosols over Qinghai‑
Tibet Plateau: Simulations with WRF‑Chem, Atmospheric Environment, 308, 119866,
https://doi.org/10.1016/j.atmosenv.2023.119866, 2023.
Zhu, C., Kanaya, Y., Takigawa, M., Ikeda, K., Tanimoto, H., Taketani, F., Miyakawa, T., Kobayashi, H.,
and Pisso, I.: FLEXPART v10.1 simulation of source contributions to Arctic black carbon, Atmos.
Chem. Phys., 20, 1641‑1656, 10.5194/acp‑20‑1641‑2020, 2020.
Zhuang, B. L., Wang, T. J., Liu, J., Li, S., Xie, M., Yang, X. Q., Fu, C. B., Sun, J. N., Yin, C. Q., Liao, J. B.,
Zhu, J. L., and Zhang, Y.: Continuous measurement of black carbon aerosol in urban Nanjing of
Yangtze     River     Delta,     China,     Atmospheric     Environment,     89,     415‑424,
https://doi.org/10.1016/j.atmosenv.2014.02.052, 2014.
Zong, Z., Wang, X., Tian, C., Chen, Y., Qu, L., Ji, L., Zhi, G., Li, J., and Zhang, G.: Source apportionment
of PM2.5 at a regional background site in North China using PMF linked with radiocarbon analysis:
insight into the contribution of biomass burning, Atmos. Chem. Phys., 16, 11249‑11265,
10.5194/acp‑16‑11249‑2016, 2016.