# Peer review of "Sources and trends of Black Carbon Aerosol in a Megacity of Nanjing,"

_EGUsphere, 2024_

## Author Comment (AC1)

**Response to Reviewer's Comments to**

Abudurexiati·Abulimiti et al, "*Sources and trends of Black Carbon Aerosol in a Megacity of Nanjing, East China After the China Clean Action Plan and Three-Year Action Plan*"

We thank the reviewers for their thoughtful and constructive comments on our paper. To guide the review process we have copied the reviewer comments in black font. Our responses are in blue. We have responded to all the referee comments and made the modification accordingly.

Anonymous Referee 1#

General comments:

Few researches have focused on the long-term changes in BC. Three-year observation data of BC in Nanjing were used to predict the changes of BC in 2014-2021 by the machine learning method. Moreover, the Aethalometer model was used to identify the source contributions of BC, including liquid fuels and solid fuels. Results revealed that the contributions of liquid fuels combustion to BC were estimated to be 80 %, which was responsible for 77% reduction of BC. The study develop a novel methodology of predicting long-term changes in BC. However, some issues need to be clarified. I would thus recommend a minor revision to improve this manuscript.

Reply: We thank the reviewer for the nice summary of our paper and the positive comments. In the following we will response to each comment listed below separately.

Minor comments:

(1) Lines 19-35, the logical coherence of the abstract needs adjustment. For instance, the conclusions of the three-year observational data of BC should be summarized first, followed by the results of the long-term predictions of BC.

Reply: Thanks for the comments. In the revised manuscript we have summarized three-year observational data at first and followed by long-term predictions of BC to improve coherence of the abstract. The corresponding sentence is changed as follows: *"Here, three-year BC observations (2019-2021) were reported in Nanjing, a polluted city in Yangtze River Delta (YRD) region, eastern China. The results revealed that the average BC concentration was 2.5 ± 1.6 µg m⁻³, peaking in winter, with approximately 80% attributed to liquid fuel combustion. Based on three-year monitoring data, the random forest (RF) algorithm was employed to reconstruct BC concentrations in Nanjing from 2014 to 2021. Source apportionment was conducted on the reconstructed time series, which revealed a significant decrease (p < 0.05) in BC levels over the eight-year period, primarily due to reduced emissions from liquid fuels.*

*Compared to the earlier control policy period (P1:2013-2017), BC concentrations declined more steeply after 2018 (P2) due to reduced solid fuel burning. The seasonal analysis indicated significant reductions (p < 0.05) in BC, $BC_{liquid}$ (black carbon from liquid fuel combustion) and $BC_{solid}$ (black carbon from solid fuel combustion) during winter, with $BC_{liquid}$ accounting for 77% of the reduction. Overall, emission reduction was the dominant factor in lowering BC levels, contributing between 62% to 86%, though meteorological conditions played an increasingly important role in P2, particularly for BC and $BC_{liquid}$. Our results demonstrate that target control measures for liquid fuel combustion are necessary, as liquid fuel combustion is a major driver for decreasing BC and highlight the non-negligible influence of meteorological factors on long-term BC variations."*

(2) Lines 167-170, here, why are 470 nm and 950 nm chosen to identify source contributions of BC? Please add the related description of this in the manuscript.

Reply: Thanks for the reviewer's comment. The choice of 470 and 950 nm wavelengths for BC source apportionment is based on several considerations. First, the absorption at 370 nm is affected by brown carbon, which introduces uncertainty into source apportionment results. Moreover, many previous studies have also used the 470 and 950 nm for BC source apportionment (Ding et al., 2023; Ding et al., 2024; Zheng et al., 2020), as it is more convenient to compare when the same wavelengths are used. Additionally, Zotter et al. (2017) have shown that source apportionment of BC at 470 nm and 950 nm were much more consistence with those using radiocarbon techniques. Consequently, the absorptions at 470 nm and 950 nm were chosen for source apportionment. We have added the related description in revised manuscript as follows: *"Considering brown carbon exhibits strong absorption at 370 nm and that BC source apportionment at 470 and 950 nm are more consistent with those using radiocarbon techniques (Zotter et al., 2017), the absorptions at 470 and 950 nm were ultimately chosen for source apportionment."*

(3) Lines 279-280, it shows that the proportion of BC to $PM_{2.5}$ can be as high as 99%. Is there a possibility of an expression error here?

Reply: Thank you for pointing out this issue. Upon reviewing the data, we discovered that the unusually high $BC/PM_{2.5}$ ratio up to 99% was due to the inclusion of outliers in the $PM_{2.5}$ data during the calculation. After removing the outliers, we found that the $BC/PM_{2.5}$ ratio now ranges from 0.6% to 26%. We have corrected the corresponding description in revised manuscript as follows: *"The hourly ratio of BC to $PM_{2.5}$ varied from 0.6 to 26%, with an annual average of 10%."* Additionally, the updated data are presented in the revised Figure 1.

[Figure]

Figure 1 (A) Hourly (dots) and daily (line) concentration of BC, PM2.5 and BC/PM2.5 and (B) frequency of BC for each year during 2019, 2020 and 2021. N represents number of hourly BC concentration for one year

Anonymous Referee 2#

General comments:

The study focuses on the long-term trends and sources of black carbon (BC) aerosol in Nanjing, China, using three years of observational data (2019–2021) combined with historical reconstruction (2014–2021) via a random forest model. Based on K-Z filter approach, it investigates the contributions of liquid and solid fuel combustion, the effectiveness of emission reduction measures, and the interplay between meteorology and emissions. The results highlight significant decreases in BC levels driven predominantly by reductions in liquid fuel combustion, with varying seasonal and meteorological influences. The findings are of significance, and I have some comments for the authors to consider.

Reply: We thank the reviewer for the nice summary of our paper and the positive comments. In the following we will response to each comment listed below separately.

Specific comments:

(1) In Figure 5, are the data presented from the training dataset or the test dataset of the random forest model? High R-square values for the training dataset could indicate overfitting if the model fails to replicate results for the test dataset, potentially compromising its generalizability. To evaluate the robustness and reliability of the random forest model, it is essential to include validation results specifically for the test dataset.

Reply: Thank you for the comment. In the original manuscript, Figure 5 presents the combined results from both the training and test datasets. As suggested by the reviewer, we have revised Figure 5 to show the results for the test dataset. The results show that the $R^2$ values for BC predictions at 370 nm and 880 nm using the test dataset are 0.90 and 0.91, respectively, which are very close to the $R^2$ values obtained using training dataset ($R^2$ range between 0.97-0.98, shown in Figure S2 and Figure S3 at specific comment 4). This indicates that the model performs consistently on unseen data, further confirming its generalizability and robustness. We have modified the corresponding sentence as follows: *"The results showed that the RF model explained over 90% of the variation in BC concentrations, with $R^2$ values of 0.90 and 0.91 between the monitored and predicted results at both 370 and 880 nm, respectively. The RF model's predictions for the test dataset were close to those for the training dataset, indicating consistent performance across both datasets and demonstrating its stability and reliability.*

[Figure]

Figure 5 Density scatter plots of hourly observed and modeled BC at (a) 370 nm and (b) 880 nm from the test dataset

(2) Line 505: The AAE values of 1 and 2 used for the Aethalometer model require justification within your study. Source apportionment outputs can vary significantly depending on the AAE values assigned for fossil fuel and biomass burning. Please provide evidence supporting the chosen values.

Reply: We thank the reviewer for pointing out the need to justify the AAE values used in the Aethalometer model. Indeed, the choice of AAE values can significantly influence BC source apportionment results. When employing the Aethalometer model, most studies typically adopted AAE values within the ranges of 0.8-1.1 for liquid fuel combustion and 1.8-2.2 for solid fuel combustion (Helin et al., 2018; Dumka et al., 2019; Fuller et al., 2014; Jing et al., 2019). To assess the uncertainty associated with the Aethalometer model, we conducted source apportionments using various AAE pairs and the results are shown in Figure S9. Our results showed that liquid fuel remained the dominant BC source in Nanjing, regardless of which AAE combination was used. Furthermore, the overall patterns of source apportionment were consistent across different AAE combinations. The uncertainty of source apportionment was calculated based on the differences between results obtained with other AAE values and those set to 1 and 2. We found that the uncertainty for $BC_{liquid}$ was 10% and for $BC_{solid}$, it was 36%. Additionally, similar AAE combinations were also used in BC source apportionment studies in Nanjing and other sites in China (Ding et al., 2024; Liu et al., 2018; Lin et al., 2021). The related description was added in revised manuscript as follows: *"It is important to highlight that the results of the Aethalometer model are highly dependent on the determination of AAE values, with $AAE_{liquid}$ ranges between 0.8 to 1.1, and $AAE_{solid}$ values ranges between 1.8 to 2.2, as widely used in this model (Helin et al., 2018; Dumka et al., 2019; Fuller et al., 2014; Jing et al., 2019). To assess the model's uncertainty, source apportionment was conducted using various AAE pairs (Figure S9). The result revealed that liquid fuel remained a dominant source of BC even*

*when different AAE paired values were used, with the pattern of source apportionment results consistent across different AAE combinations. The AAE$_{liquid}$ =1 and AAE$_{solid}$=2 were used in this study, as the same combination of AAE values were utilized in Nanjing and other sites in China (Ding et al., 2024; Liu et al., 2018; Lin et al., 2021). Additionally, the uncertainty of source apportionment was estimated based on the relative differences between results obtained with other AAE values and those set to 1 and 2. As a result, the uncertainty of the BC$_{lqiuid}$ was estimated to be 10%."*

[Figure]

Figure S9 Uncertainty analysis of BC source apportionment using the Aethalometer model. Contributions of liquid and solid fuel combustion to BC were calculated using five different combinations of AAE$_{liquid}$ and AAE$_{solid}$

(3) Precipitation is an essential parameter for BC scavenging, yet it appears to be absent from the input variables in your random forest model and should be included.

Reply: Thank you for your comment. Precipitation plays an important role in black carbon wet scavenging process (Ding et al., 2023; Ding et al., 2024; Liu et al., 2020). As suggested by the reviewer, we have considered hourly precipitation as an input variable in the random forest model. However, the variable importance analysis revealed that precipitation had the lowest contribution to the model's predictive performance (Figure I). This is likely because the impact of precipitation on black carbon typically accumulates over a long-term scale, while the model input was on an hourly scale, which may not fully capture the long-term effects of precipitation. Additionally, the comparison of model predictions with and without precipitation showed that the $R^2$ between the two sets of predictions was as high as 0.99, with no significant ($p = 0.50$) difference in their averages (Figure II). This suggests that the inclusion of precipitation had little effect on the model's performance. Based on these results, we decided to exclude precipitation from the model. And the related description was added in revised manuscript as

follows: " *plays a key role in the wet scavenging of BC (Liu et al., 2020; Ding et al., 2024), its inclusion in the RF model showed a minimal contribution to predicting BC concentrations. The relatively lower contribution of precipitation can be attributed to the fact that its impact on BC typically appears over a longer time scale, while the model input is based on hourly precipitation, which may not adequately capture the cumulative. Furthermore, including precipitation in the model had no significant impact on its predictive performance. Thus, precipitation was excluded from the RF model.*"

[Figure]

Figure I The predictor's importance for BC at (A) exclude precipitation and (B) include precipitation

[Figure]

Figure II Comparison of model predictions with and without precipitation: (A) Comparison of predicted BC concentration, (B) Difference in predicted BC concentration

(4) Lines 212–215. Please present figures showing the results of the 10-fold CV used for the random forest model. These figures would allow reviewers to assess the model-building process and ensure robustness. Such details could be included in the supplementary materials.

Response: Thank you for the comment. In the revised manuscript, we have added new figures

(Figure S2 and Figure S3) showing the results of the 10-fold cross-validation (CV) for the random forest model in the Supporting Information. The model showed consistent performance across all folds, with only minimal variation in $R^2$ values (0.98 for BC at 370 nm and 0.97-0.98 for BC at 880 nm). MAE values for BC at 370 nm ranged from 0.37 to 0.41, and for BC at 880 nm from 0.29 to 0.30. The RMSE values were between 0.57 and 0.74 for BC at 370 nm and between 0.47 and 0.54 for BC at 880 nm, showing minimal variation as well. These results confirmed the stability and reliability of the model during the training process. The related description was added in revised manuscript as follows: *"The results showed that the RF-predicted BC at 880 nm correlated well with the observations, with an average $R^2$ of 0.97, MAE varying from 0.29 to 0.30, and RMSE ranging from 0.47 to 0.54. For BC at 370 nm, the cross-validation results were also robust, with a mean $R^2$ of 0.98, MAE values ranging from 0.37 to 0.41, and RMSE values varying from 0.57 to 0.74, which confirms the stability and reliability of the model."*

[Figure]

Figure S2 10-fold cross-validation results for the random forest model predicting BC concentration at 880 nm. Each panel in the figure corresponds to one of the 10 folds

[Figure]

Figure S3 10-fold cross-validation results for the random forest model predicting BC concentration at 370 nm. Each panel in the figure corresponds to one of the 10 folds

(5) A figure illustrating the K-Z filter results, showing the long-term trends for emissions and meteorology, can enhance clarity. Please include. Additionally, when discussing the trends of pollutants, consider applying the Mann-Kendall test to assess the statistical significance of these trends.

Response: Thanks for the reviewer's comment. In the revised manuscript, we have added a new Figure S10 in the Supplementary Materials that shows the KZ filter results. The trends for emission-related components of BC, $BC_{liquid}$ and $BC_{solid}$ were similar, remaining stable until 2017, then peaking in 2019, and then sharply declining by the end of 2020. In contrast, meteorology-related trends showed a sharp decrease after 2020 for BC and $BC_{liquid}$, while $BC_{solid}$ showed a steady downward trend over the entire period (Figure S11). To make it clear, the sentence was added in revised manuscript as follows: *"The emission-related components of BC, $BC_{liquid}$ and $BC_{solid}$ exhibited similar long-term trends (Figure S11). From 2014 to 2016, the emission-related trends remained relatively stable, reaching a lower level by the end of 2017.Subsequently, the emission-related components of BC, $BC_{liquid}$ and $BC_{solid}$ increased, peaking in 2019, followed by a sharp decline until mid-2020, and then rebounding to another peak at the end of 2021, which may be related to the recovery of production activities following the pandamic. In contrast, meteorology-related trends of BC and $BC_{liquid}$ showed a sharp decrease after 2020, while $BC_{soild}$ exhibited a downward trend between 2014 and 2021, with meteorology-related trends of $BC_{solid}$ followed a fluctuating downward pattern."*

Additionally, we applied the Mann-Kendall test to air pollutants during the study period and the results were shown in Table 4 and Table S4. Compared to the multiple linear regression results, the interannual trends of pollutants remained unchanged, with only the significance intervals shifting. Similarly, the seasonal trends of pollutants showed no major differences; however, the slopes estimated by Sen's slope were slightly higher than those obtained by multiple linear regression. The only notable difference in pollutant trends was that the change in $NO_2$ during autumn was not statistically significant.

[Figure]

Figure S11 (A) Emission-related and (B) meteorology-related trends of BC. The left, middle and right panels represent BC, $BC_{liquid}$ and $BC_{solid}$

Table 4 The change rates of BC and other air pollutants during different period

| Study period | air pollutants | absolute slope[a] | relative slope[b] | p |
|---|---|---|---|---|
| Air Pollution Prevention and Control Action Plan | BC | -0.12 | -4.18% | 0.08 |
| | $BC_{liquid}$ | -0.10 | -4.26% | 0.02 |
| | $BC_{solid}$ | -0.02 | -3.48% | 0.6 |
| | $PM_{2.5}$ | -12.00 | -26.29% | 0.0001 |
| | $NO_2$ | -0.46 | -1.26% | 0.74 |
| | $SO_2$ | -1.69 | -10.08% | 0.06 |
| | CO | 0.02 | 1.76% | 0.62 |
| After 2018 | BC | -0.29 | -11.22% | 0.0002 |
| | $BC_{liquid}$ | -0.21 | -10.26% | 0.0001 |
| | $BC_{solid}$ | -0.05 | -11.55% | 0.06 |
| | $PM_{2.5}$ | -4.62 | -17.20% | 0.0009 |
| | $NO_2$ | -2.91 | -8.73% | 0.02 |
| | $SO_2$ | -2.32 | -33.23% | 0.0001 |
| | CO | 0.00 | 0.00% | 0.66 |

[a]: $\mu g\ m^{-3}\ yr^{-1}$

[b]: % yr-1

**Table S4 The change rates of BC and other air pollutants across different seasons**

| season | pollutants | absolute slope[a] | relative slope[b] | p |
|--------|-----------|-------------------|-------------------|-----|
| spring | BC | -0.19 | -7.16% | 0.003 |
| | $BC_{liquid}$ | -0.16 | -7.60% | 0.003 |
| | $BC_{solid}$ | -0.05 | -8.74% | 0.013 |
| | $PM_{2.5}$ | -6.00 | -15.90% | 0.001 |
| | $NO_2$ | -3.00 | -7.82% | 0.003 |
| | $SO_2$ | -3.38 | -28.15% | 0.001 |
| | CO | -0.09 | -11.67% | 0.008 |
| summer | BC | -0.13 | -5.16% | 0.013 |
| | $BC_{liquid}$ | -0.09 | -4.16% | 0.011 |
| | $BC_{solid}$ | -0.04 | -12.65% | 0.000 |
| | $PM_{2.5}$ | -6.67 | -25.44% | 0.001 |
| | $NO_2$ | -1.50 | -5.50% | 0.054 |
| | $SO_2$ | -2.20 | -20.17% | 0.000 |
| | CO | -0.04 | -6.06% | 0.090 |
| autumn | BC | -0.13 | -4.87% | 0.030 |
| | $BC_{liquid}$ | -0.11 | -5.08% | 0.008 |
| | $BC_{solid}$ | -0.01 | -1.64% | 0.790 |
| | $PM_{2.5}$ | -6.00 | -20.02% | 0.001 |
| | $NO_2$ | -0.36 | -3.28% | 0.790 |
| | $SO_2$ | -2.29 | -20.61% | 0.001 |
| | CO | -0.03 | -3.55% | 0.001 |
| winter | BC | -0.33 | -9.95% | 0.001 |
| | $BC_{liquid}$ | -0.26 | -10.50% | 0.001 |
| | $BC_{solid}$ | -0.06 | -8.55% | 0.006 |
| | $PM_{2.5}$ | -7.06 | -14.20% | 0.000 |
| | $NO_2$ | -1.33 | -3.35% | 0.413 |
| | $SO_2$ | -4.67 | -36.15% | 0.001 |
| | CO | -0.09 | -8.63% | 0.001 |

[a]: $\mu g\ m^{-3}\ yr^{-1}$

[b]: % yr-1

(6) Provide an explanation for the differing impacts of meteorology on the extent of reductions in BC liquid and BC solid by the K-Z method.

Response: $BC_{liquid}$ emissions remain relatively stable throughout the year, making it more sensitive to meteorological conditions. In contrast, $BC_{solid}$ emissions show significant seasonal variation. Seasonal significance analysis of meteorology-related $BC_{liquid}$ and $BC_{solid}$ obtained by the KZ filter method (Figure S13). The results revealed no significant differences in spring and summer. However, in the autumn and winter, when biomass burning and coal combustion are more intense in China, meteorological conditions had significantly different impacts on $BC_{liquid}$ and $BC_{solid}$ ($p < 0.05$). This seasonal variability in emissions explains why meteorological conditions have different effects on $BC_{liquid}$ and $BC_{solid}$. To make it clear, the sentence was added in revised manuscript as follows: "*It is worth noting that the impact of meteorological conditions on $BC_{liquid}$ and $BC_{solid}$ differs significantly, especially in P2. While meteorology contributed 70% to the reduction in $BC_{liquid}$, its impact on $BC_{solid}$ was only 31%. This difference is because $BC_{liquid}$, mainly from vehicle exhaust, remains stable year-round, whereas $BC_{solid}$, from activities like biomass burning and coal combustion, varies seasonally. The results of significance analysis further confirmed that there was no significant difference in $BC_{liquid}$ and $BC_{solid}$ during spring and summer while significant ($p < 0.05$) differences were observed in autumn and winter, when BCsolid emissions are more pronounced due to increased biomass burning and coal combustion activities (Figure S13). This seasonal variability in emission sources explains the differing impacts of meteorology on $BC_{liquid}$ and $BC_{solid}$.*"

[Figure]

Figure S13 Seasonal variation of meteorology-related $BC_{liquid}$ and $BC_{solid}$. The square in the figure represents the 25th and 75th percentiles of the data, the vertical lines represent the 10th and 90th percentiles, and the horizontal line inside the square indicates the median. The Y-axis represents the log-transformed concentrations of $BC_{liquid}$ and $BC_{solid}$

(7) Extend the comparison of your findings to other regions in China and globally. In China,

BC emissions exhibit notable geographic heterogeneity. Beyond the commonly studied regions like the North China Plain and East China, comparisons to other areas would be beneficial. For example, studies conducted in Nanning, Guangxi province (DOI: 10.1016/j.scitotenv.2023.166747) and in Liaoning province (DOI: 10.1016/j.envpol.2024.124470) can offer valuable context. Similarly, discussing the BC/CO and BC/PM$_{2.5}$ ratios in your study relative to that in these works can enhance the scientific depth of your analysis.

Response: We agreed that extending the comparison of our findings to other regions in China and globally can provide valuable insights for understanding geographic heterogeneity in BC emissions. Therefore, we have added BC concentrations from Benxi and Nanning in Table 1 to compare BC levels across China. Furthermore, we conducted a trend analysis of the BC/PM$_{2.5}$ and BC/CO ratios across the whole study period (Figure S9). The BC/PM$_{2.5}$ ratio showed a significant increasing trend ($p < 0.01$), indicating that while emission reduction policies have been effective in decreasing secondary aerosol precursors (SO$_4^{2-}$, NO$_3^-$, NH$_4^+$), more stringent regulations on BC emissions may be needed. We also considered the BC/CO ratio; however, no significant trend was observed. The related description was added as follows: *"Pollutants commonly co-emitted with BC, such as NO₂, CO, and SO₂, exhibited significant declining trends ($p < 0.05$) during the study period (Figure S9). In contrast, the BC/PM$_{2.5}$ ratio showed a significant increasing trend ($p < 0.01$), suggesting that while emission reduction policies have been effective in decreasing precursors of secondary aerosol (SO$_4^{2-}$, NO$_3^-$, NH$_4^+$), stricter regulations on BC emission may also be necessary. The variation in the BC/CO ratio was not significant, with the mean value remaining stable at approximately 0.38% throughout the whole period."*

[Figure]

Figure S9 Trends in air pollutants at sampling site. The solid black line represents the monthly

medians, the dash black lines represent the 10th and 90th monthly percentiles, and the orange line is the fitted long-term trend.

(8) Some typos and grammar need to be corrected, such as line 71 "severe", line 120-121. "was then incorporated", Line 318, "Similarly". Thoroughly check the manuscript for similar errors to ensure clarity and precision.

Response: Thank you for your thorough review of our manuscript. We appreciate your attention to detail and have thoroughly reviewed the manuscript for typos and grammatical errors. The imprecise expressions in the manuscript have been revised accordingly.

References:

Ding, S., Zhao, D., Tian, P., and Huang, M.: Source apportionment and wet scavenging ability of atmospheric black carbon during haze in Northeast China, Environmental Pollution, 357, 124470, https://doi.org/10.1016/j.envpol.2024.124470, 2024.

Ding, S., Liu, D., Zhao, D., Tian, P., Huang, M., and Ding, D.: Characteristics of atmospheric black carbon and its wet scavenging in Nanning, South China, Science of The Total Environment, 904, 166747, https://doi.org/10.1016/j.scitotenv.2023.166747, 2023.

Dumka, U. C., Kaskaoutis, D. G., Devara, P. C. S., Kumar, R., Kumar, S., Tiwari, S., Gerasopoulos, E., and Mihalopoulos, N.: Year-long variability of the fossil fuel and wood burning black carbon components at a rural site in southern Delhi outskirts, Atmospheric Research, 216, 11-25, https://doi.org/10.1016/j.atmosres.2018.09.016, 2019.

Fuller, G. W., Tremper, A. H., Baker, T. D., Yttri, K. E., and Butterfield, D.: Contribution of wood burning to PM10 in London, Atmospheric Environment, 87, 87-94, https://doi.org/10.1016/j.atmosenv.2013.12.037, 2014.

Helin, A., Niemi, J. V., Virkkula, A., Pirjola, L., Teinilä, K., Backman, J., Aurela, M., Saarikoski, S., Rönkkö, T., Asmi, E., and Timonen, H.: Characteristics and source apportionment of black carbon in the Helsinki metropolitan area, Finland, Atmospheric Environment, 190, 87-98, https://doi.org/10.1016/j.atmosenv.2018.07.022, 2018.

Jing, A., Zhu, B., Wang, H., Yu, X., An, J., and Kang, H.: Source apportionment of black carbon in different seasons in the northern suburb of Nanjing, China, Atmospheric Environment, 201, 190-200, https://doi.org/10.1016/j.atmosenv.2018.12.060, 2019.

Lin, Y.-C., Zhang, Y.-L., Xie, F., Fan, M.-Y., and Liu, X.: Substantial decreases of light absorption, concentrations and relative contributions of fossil fuel to light-absorbing carbonaceous aerosols attributed to the COVID-19 lockdown in east China, Environmental Pollution, 275, 116615, https://doi.org/10.1016/j.envpol.2021.116615, 2021.

Liu, D., Ding, S., Zhao, D., Hu, K., Yu, C., Hu, D., Wu, Y., Zhou, C., Tian, P., Liu, Q., Wu, Y., Zhang, J., Kong, S., Huang, M., and Ding, D.: Black Carbon Emission and Wet Scavenging From Surface to the Top of Boundary Layer Over Beijing Region, Journal of Geophysical Research: Atmospheres, 125, e2020JD033096, https://doi.org/10.1029/2020JD033096, 2020.

Liu, Y., Yan, C., and Zheng, M.: Source apportionment of black carbon during winter in Beijing, Science of The Total Environment, 618, 531-541,

https://doi.org/10.1016/j.scitotenv.2017.11.053, 2018.

Zheng, H., Kong, S., Zheng, M., Yan, Y., Yao, L., Zheng, S., Yan, Q., Wu, J., Cheng, Y., Chen, N., Bai, Y., Zhao, T., Liu, D., Zhao, D., and Qi, S.: A 5.5-year observations of black carbon aerosol at a megacity in Central China: Levels, sources, and variation trends, Atmospheric Environment, 232, 117581, https://doi.org/10.1016/j.atmosenv.2020.117581, 2020.

Zotter, P., Herich, H., Gysel, M., El-Haddad, I., Zhang, Y., Močnik, G., Hüglin, C., Baltensperger, U., Szidat, S., and Prévôt, A. S. H.: Evaluation of the absorption Ångström exponents for traffic and wood burning in the Aethalometer-based source apportionment using radiocarbon measurements of ambient aerosol, Atmos. Chem. Phys., 17, 4229-4249, 10.5194/acp-17-4229-2017, 2017.